# VIDEO-AS-PROMPT: UNIFIED SEMANTIC CONTROL FOR VIDEO GENERATION

**Yuxuan Bian**[1,2*], **Xin Chen**[2†], **Zenan Li**[2], **Tiancheng Zhi**[2], **Shen Sang**[2], **Linjie Luo**[2‡], **Qiang Xu**[1,3‡]

[1]The Chinese University of Hong Kong (CUHK) [2]ByteDance [3]Shenzhen Loop Area Institute

{yxbian,qxu}@cse.cuhk.edu.hk {chris.chen1,linjie.luo}@bytedance.com

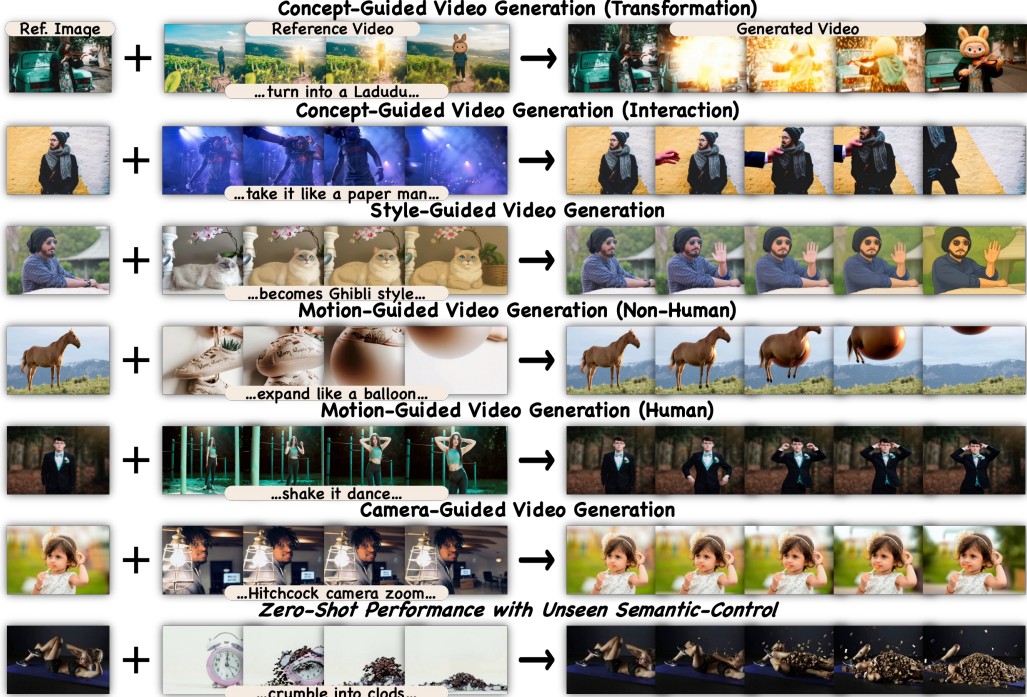

Figure 1: **Video-As-Prompt (*VAP*) is a unified semantic-controlled video generation framework: it treats *reference videos with wanted semantics as video prompts* and controls generation via *a plug-and-play, in-context Mixture-of-Transformers expert*.** *Row* 1−6: reference videos used as prompts for diverse semantic-controlled tasks (concept, style, motion, camera). *Row* 7: zero-shot results from *VAP* when given an unseen semantic, demonstrating strong generalization. **We strongly encourage readers to view our project page**[1]**for better visualization.**

## ABSTRACT

Unified, generalizable semantic control in video generation remains a critical open challenge. Existing methods either introduce artifacts by enforcing inappropriate pixel-wise priors from structure-based controls, or rely on non-generalizable, condition-specific finetuning or task-specific architectures. We introduce **Video-As-Prompt (VAP)**, a new paradigm that reframes this problem as in-context generation. VAP leverages a reference video as a direct semantic prompt, guiding a frozen Video Diffusion Transformer (DiT) via a plug-and-play Mixture-of-Transformers (MoT) expert. This architecture prevents catastrophic forgetting and is guided by a temporally biased position embedding that eliminates spurious mapping priors for robust context retrieval. To power this approach and catalyze future research, we built **VAP-Data**, the largest dataset for this task with over

---

*This work was done when Yuxuan Bian was an intern at ByteDance.

[†]Project Lead.

[1]Our project page is at https://bytedance.github.io/Video-As-Prompt/.

$100K$ paired videos across 100 semantic conditions. As a single unified model, VAP sets a new state-of-the-art for open-source methods, achieving a **38.7% user preference rate** that rivals leading condition-specific commercial models. VAP's strong zero-shot generalization and support for various applications mark a significant advance toward general-purpose, controllable video generation.

# 1 INTRODUCTION

While unified structure-controlled video generation (Jiang et al., 2025) under pixel-aligned conditions (*e.g.*, depth (Gao et al., 2025a), pose (Hu, 2024), mask (Bian et al., 2025b), optical flow (Jin et al., 2025)) is well studied, semantic-controlled generation—lacking a pixel-aligned condition (*e.g.*, concept (Liu et al., 2025), style (Ye et al., 2025), motion (Zhang et al., 2025b), camera (Bai et al., 2025a)) to the target video—remains fragmented without a unified and generalizable framework, limiting applications in visual effects (Macklin et al., 2014), video stylization (Jamriška et al., 2019), motion imitation (Wang et al., 2025), and camera control (Bai et al., 2025b).

Migrating current unified structure-controlled methods (Jiang et al., 2025; Zhang et al., 2023) often causes artifacts because they enforce inappropriate pixel-wise mapping priors from structure-based control abilities (see Fig. 2 (a)). Other semantic-controlled methods fall into two groups: (1) **Condition-Specific Overfit** (see Fig. 2 (b)): methods (Liu et al., 2025; Civitai, 2025) finetune backbones (Yang et al., 2024; Wan et al., 2025) or LoRAs (Hu et al., 2022) for each semantic condition (*e.g.*, Ghibli style, Hitchcock camera zoom), which is costly; (2) **Task-Specific Design** (see Fig. 2 (c)): methods (Ye et al., 2025; Bai et al., 2025a; Zhang et al., 2025b) craft task-specific modules or inference strategies for a condition type (*e.g.*, style, camera), often encoding videos with the same semantics to a specially designed space and guiding generation. While effective, these condition/task-specific approaches hinder a unified model and limit zero-shot generalizability.

However, recent image generation (Tan et al., 2025) and structure-controlled video generation (Ju et al., 2025) show that Diffusion Transformers (DiTs) support strong in-context control, motivating a unified framework for in-context semantic-controlled video generation. As shown in Fig. 2 (d), rather than assuming pixel-wise correspondence (Jiang et al., 2025), training per-condition models (Liu et al., 2025) or using task-specific designs (Ye et al., 2025), we treat the video of the wanted semantics as a reference video prompt and guide generation via in-context control. This formulation removes the inappropriate pixel-wise mapping prior from structure-based controls, avoids per-condition training or per-task model designs, and enables a single unified model to handle diverse semantic controls and generalize in a zero-shot manner to unseen semantics (see Fig. 1).

We present **V**ideo-**A**s-**P**rompt (*VAP*), the first unified framework for semantic-controlled video generation under non-pixel-aligned conditions, by treating a reference video with the wanted semantics as a video prompt and using plug-and-play in-context control. As shown in Fig. 2 (d), *VAP* adopts a plug-and-play Mixture-of-Transformers (MoTs) design (Liang et al., 2024) to augment any frozen Video Diffusion Transformer (Peebles & Xie, 2023) with a trainable parallel expert for interpreting the video prompt and guiding the generation, preventing catastrophic forgetting and enabling in-context control. The expert (for the reference prompt) and the frozen backbone (for target generation) run independent feed-forward and layer-norm paths and communicate via full attention for synchronous layer-wise reference guidance. For robust context retrieval, we adopt a temporally biased Rotary Position Embedding (RoPE) that places the reference before the current video along the temporal axis while keeping spatial fixed; this removes the nonexistent pixel-mapping prior from a shared RoPE, matches the temporal order expected by in-context generation, and preserves spatial consistency so the model can exploit spatial semantic changes of the reference video prompt.

Existing datasets (Jiang et al., 2025; Ju et al., 2025) lack focus on semantic-controlled video generation. We introduce *VAP-Data*, the largest dataset to date, with over $100K$ curated samples across 100 semantic conditions, providing a robust data foundation for unified semantic-controlled video generation. Extensive experiments show that *VAP*, a unified model for diverse semantic conditions (Sec. A) and downstream generation tasks (Sec. B), produces coherent, semantically aligned videos, achieves a 38.7% user preference rate competitive with leading closed-source commercial models, surpasses condition-specific methods, and exhibits zero-shot generalization (Fig. 7).

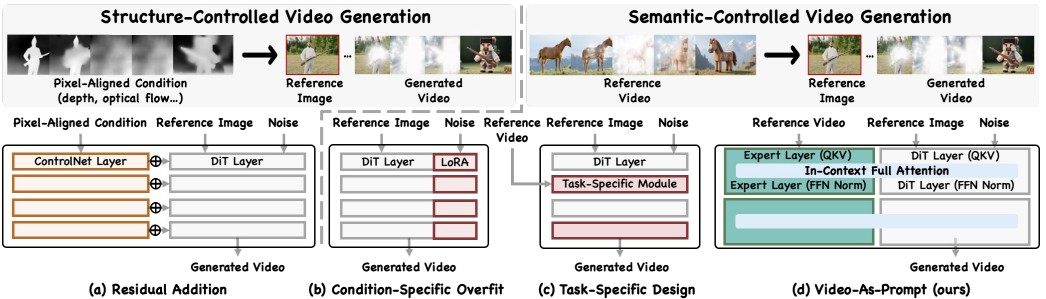

Figure 2: **Controllable Video Generation Paradigms. Structure-Controlled Video Generation (a)**. The condition is pixel-aligned with the target video. Most works inject conditions (*e.g.*, depth, optical flow) into a DiT via an extra branch using *(a) residual addition*, leveraging pixel-wise alignment. **Semantic-Controlled Video Generation (b, c, d)**. The condition and target video share the same semantics. Most works use *(b) Condition-Specific Overfit* or *(c) Task-Specific Design*: fine-tuning per semantic or adding task-specific modules. *(d) Video-as-Prompt*: We use a reference video with the same semantics as prompts and adopt a plug-and-play in-context control framework built on mixture-of-transformers to achieve unified semantic-controlled video generation.

Our contributions are highlighted as follows:

⇒ We present *VAP*, a unified semantic-controlled video generation paradigm, treating a reference video with the wanted semantics as a video prompt for generalizable in-context control.

⇒ We propose a plug-and-play in-context video generation framework built on the mixture-of-transformers architecture that prevents catastrophic forgetting, supports various downstream tasks, and delivers strong zero-shot generalization to unseen semantic condition categories.

⇒ We construct and release *VAP-Data*, the largest dataset for semantic-controlled video generation, with over $100K$ curated paired samples across 100 condition categories.

## 2 RELATED WORKS

### 2.1 VIDEO GENERATION

Video generation has progressed from early GAN-based models (Vondrick et al., 2016; Skorokhodov et al., 2022) to modern diffusion models (Brooks et al., 2024; Gao et al., 2025b). Leveraging the scalability of diffusion transformers (DiTs) (Peebles & Xie, 2023), research has moved from convolutional architectures (Singer et al., 2022; Blattmann et al., 2023; Girdhar et al., 2023; Chen et al., 2024b; Zhang et al., 2024) to transformer-based ones (Gupta et al., 2023; Menapace et al., 2024; Brooks et al., 2024; Polyak et al., 2024; Kong et al., 2024; Wan et al., 2025; Gao et al., 2025b). The standard pipeline encodes Gaussian noise into a latent space with a VAE (Kingma & Welling, 2013), splits the latents into patches, processes the patches with a DiT, and decodes to pixel space to produce high-quality, smooth videos. However, pre-trained DiTs typically support only text prompts or first/last-frame control (Wan et al., 2025; Gao et al., 2025b). To enable finer, user-defined control, many methods add task-specific modules to pre-trained DiTs (Jiang et al., 2025; Bian et al., 2025b) or design special inference (Zhang et al., 2025b; Wang et al., 2025) for new controllable video tasks.

### 2.2 CONTROLLABLE VIDEO GENERATION

In general, controllable video generation can be categorized into **Structure-Controlled Video Generation** and **Semantic-Controlled Video Generation** (see top of Fig. 2). The former (Jiang et al., 2025; Bian et al., 2025b) is driven by pixel-aligned conditions (*e.g.*, depth, pose, mask, optical flow), while the latter (Zhang et al., 2025b; Ye et al., 2025) focuses on generation based on semantic conditions without pixel mapping prior (*e.g.*, concept, style, motion, camera).

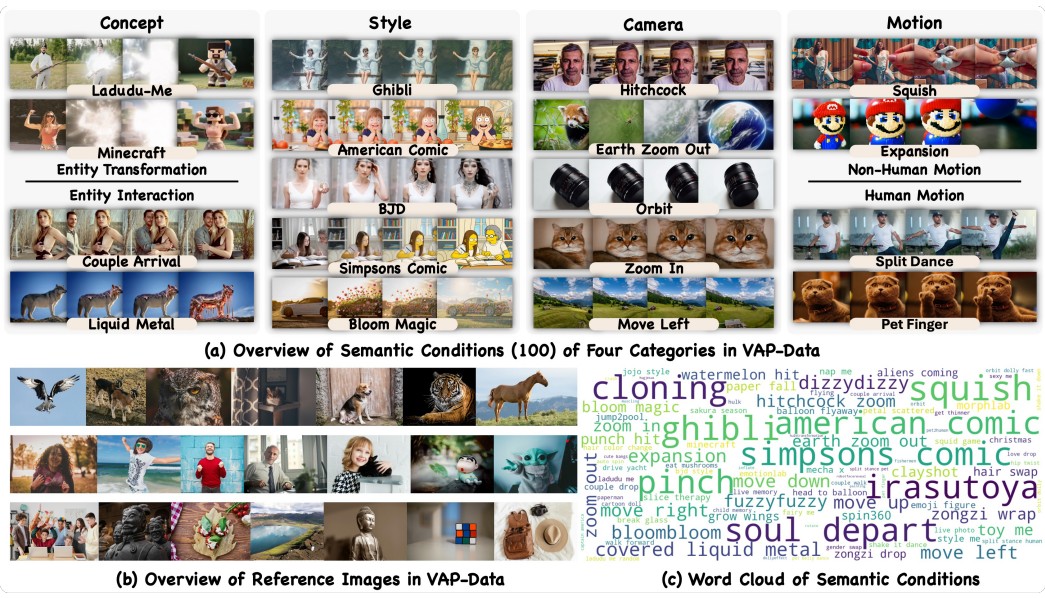

Figure 3: **Overview of Our Proposed *VAP-Data*.** (a) 100 semantic conditions across 4 categories: concept, style, camera, and motion; (b) diverse reference images, including animals, humans, objects, and scenes, with multiple variants; and (c) a word cloud of the semantic conditions.

**Structure-Controlled Video Generation** In structure-controlled video generation, condition videos (*e.g.*, depth, pose) are typically pixel-aligned with the target videos, so control signals are mostly modeled with an additional adapter/branch and injected via residual addition to exploit this mapping prior (Zhang et al., 2023; Mou et al., 2024; Bian et al., 2025a; Lin et al., 2024), as shown in Fig. 2 (a). Common conditions include **Trajectory** (Wang et al., 2024; Zhou et al., 2024; Guo et al., 2025a; Gu et al., 2025), **Pose** (Ju et al., 2023; Lei et al., 2024; Hu, 2024), **Depth** (Esser et al., 2023; Gao et al., 2025a), **Optical flow** (Jin et al., 2025; Koroglu et al., 2025), and **Mask** (Bian et al., 2025b; Yang et al., 2025). Recent works (Ju et al., 2025; Jiang et al., 2025) further enable all-in-one structure-controlled generation by treating these inputs as a unified, pixel-aligned spatial condition.

**Semantic-Controlled Video Generation** Semantic-controlled video generation handles conditions which lack pixel-wise correspondence with target videos (see Fig. 1), including **Concept** (Liu et al., 2025; Hsu et al., 2024; Pika, 2025; PixVerse, 2025; Vidu, 2025; Kling, 2025) (*e.g.*, turning an object into Ladudu or taking it like a paper man), **Stylization** (Huang et al., 2025; Ye et al., 2025), **Camera Movement** (He et al., 2024; Bai et al., 2025a;b), and **Motion** (Zhao et al., 2024; Yatim et al., 2024; Pondaven et al., 2025; Zhang et al., 2025b; Wang et al., 2025), where the reference and target share motion but differ in layout or skeleton. As shown in Fig. 2 (b) and (c), prior methods fall into **Condition-Specific Overfit** (Liu et al., 2025; Civitai, 2025), which fine-tune DiT backbones or LoRAs for each semantic condition; and **Task-Specific Design** (Ye et al., 2025; Bai et al., 2025a; Zhang et al., 2025b; Wang et al., 2025), which add task-specific modules or inference strategies for a class of semantic conditions (*e.g.*, style, motion, camera). Above approaches fit narrow distributions but are not unified and generalizable; they require per-condition retraining or per-task designs and lack zero-shot generalization. A concurrent work (Mao et al., 2025) adopts a LoRA mixture-of-experts for unified generation across multiple semantic conditions, but it still learns each condition by overfitting subsets of parameters and fails to generalize to unseen ones. This raises a key question: **How can we build a unified semantic-controlled video generation framework?**

Inspired by in-context learning (Huang et al., 2024; Bian et al., 2024; Tan et al., 2025; Guo et al., 2025b; Ju et al., 2025), we propose Video-As-Prompt (*VAP*), which treats videos with the wanted semantics as unified in-context prompts to guide generation. By casting the task as an in-context generation with reference video prompts, *VAP*, to our knowledge, is the first to unify multiple semantic-controlled tasks without task-specific designs, while achieving strong zero-shot abilities.

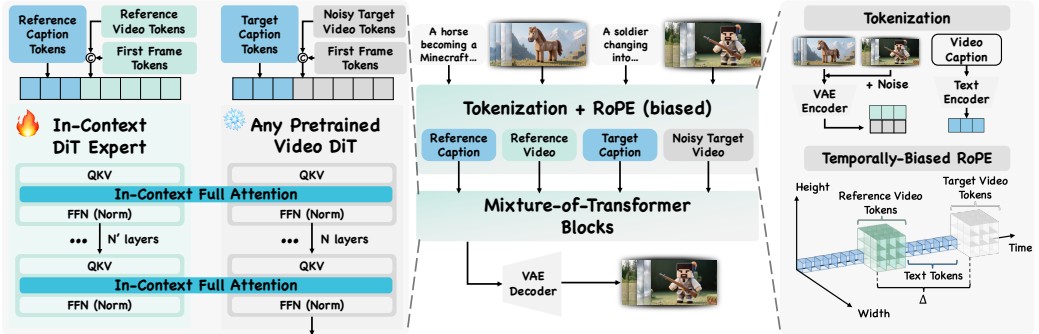

Figure 4: **Overview of Video-As-Prompt.** The reference video (with the wanted semantics), target video, and their first frames (reference images) are encoded into latents by the VAE and, together with captions (see top right), form an in-context token sequence $[Ref_{text}, Ref_{video}, Tar_{text}, Tar_{video}]$ (see middle). First frame tokens are concatenated with video tokens. We add a temporal bias $\Delta$ to RoPE to avoid nonexistent pixel-aligned priors from the original shared RoPE (see bottom right). The reference video and captions act as the prompts and are fed into a trainable DiT Expert Transformer (see left), which exchanges information bidirectionally with the pre-trained DiT via full attention at each layer, enabling plug-and-play in-context generation.

## 3 METHODS

*VAP* supports unified semantic-controlled video generation under various semantic conditions (*e.g.*, concept, style, motion, and camera). Our insight is to use videos with the wanted semantics as unified prompts to guide generation across tasks, avoiding per-condition fine-tuning or per-task designs. Although we study a limited set of conditions, the method extends to others without major structural changes and shows promising generalizability for different semantic conditions (see Sec. A), various downstream tasks (see Sec. B), and unseen semantics in training (see Fig. 7).

### 3.1 PRELIMINARY

Video diffusion models (Brooks et al., 2024; Gao et al., 2025b) learn the conditional distribution $p(\mathbf{x} \mid C)$ of video $\mathbf{x}$ given conditions $C$. Using Flow Matching (Lipman et al., 2022) for illustration, a noise sample $\mathbf{x_0} \sim \mathcal{N}(0,1)$ is denoised to $\mathbf{x_1}$ along the path $\mathbf{x_t} = t\mathbf{x_1} + (1 - (1 - \sigma_{min})t)\mathbf{x_0}$, where $\sigma_{min} = 10^{-5}$ and $t \in [0,1]$. The model $u$ is trained to predict the velocity $V_t = \frac{d\mathbf{x_t}}{dt}$, which simplifies to: $V_t = \frac{d\mathbf{x_t}}{dt} = \mathbf{x_1} - (1 - \sigma_{min})\mathbf{x_0}$. We optimize $u$ with parameters $\Theta$ by minimizing the mean squared error loss $\mathcal{L}$ between the ground-truth velocity and the model prediction:

$$\mathcal{L} = \mathbb{E}_{t,\mathbf{x_0},\mathbf{x_1},C} \left\| u_\Theta(\mathbf{x_t}, t, C) - (\mathbf{x_1} - (1 - \sigma_{min})\mathbf{x_0}) \right\|$$

During inference, the model first samples Gaussian noise $\mathbf{x_0} \sim \mathcal{N}(0,1)$ and then uses an ODE solver with a discrete set of $N$ denoising timesteps to produce $\mathbf{x_1}$.

### 3.2 REFERENCE VIDEOS AS TASK-AGNOSTIC PROMPTS

Semantic-controlled video generation spans diverse condition types (*e.g.*, concept, style, motion, camera). Structure-controlled methods assume pixel-wise alignment between condition and target (Zhang et al., 2023; Jiang et al., 2025); injecting a semantic-same but pixel-misaligned video condition via residual addition yields copy-and-paste artifacts (see Fig. 5 (a)). Prior semantic-controlled video generation works partially tackle this by using per-condition fine-tuning or per-task designs, treating tasks in isolation. In contrast, *VAP* employs *reference videos as video prompts*, which share the same semantics as the targets, independent of task category, unifying heterogeneous conditions in one model. Formally, let $\mathcal{C} = \bigcup_{i=1}^{n} C_i$ denote $n$ condition types with conditions $c \in C_i$ (total $m$); prior methods often fine-tune $n$ (per-task) or up to $m$ (per-condition) models, whereas we train a single unified model $u_\Theta$ that jointly learns $p(\mathbf{x} \mid c)$ for any $c \in \mathcal{C}$. We evaluate four represen-

Figure 5: **Motivation.** Ablation visualizations (Semantic: Spin 360°) on structure designs of *VAP*.

tative types—concept ($C_{co}$), style ($C_s$), motion ($C_m$), and camera ($C_{ca}$)—chosen for distinct task definitions and data. Our *VAP-Data* follows this taxonomy; overviews appear in Fig. 3.

- **Concept-Guided Generation**: Videos sharing a concept, such as entity transformation (*e.g.*, a person becomes a Ladudu doll) or interaction (*e.g.*, an AI lover approaches the target).
- **Style-Guided Generation**: Videos in a reference style (*e.g.*, Ghibli, Minecraft).
- **Motion-Guided Generation**: Videos following a reference motion, including non-human motion (*e.g.*, objects expand like balloons) and human motion (*e.g.*, shake it dance).
- **Camera-Guided Generation**: Videos following reference camera motion, from basic translations (up, down, left, right, zoom in/out) to the Hitchcock dolly zoom.

**Discussion.** We also input captions ($P_{ref}, P_{tar}$) of the reference video and target video to aid in finding and transferring the shared mentioned semantic control signals (*e.g.*, "cover liquid metal" in Fig. 6). Thus $u_\Theta$ learns conditional distribution $p(\mathbf{x} \mid C_{co}, C_s, C_m, C_{ca}, P_{ref}, P_{tar})$.

### 3.3 Plug-and-Play In-Context Control

Our model takes four primary inputs: a reference video (providing the semantics), a reference image[2] (providing the initial appearance and subject), captions (helping find the target semantic), and noise (inference) or noisy target video (training). We first encode the reference video $c \in \mathbb{R}^{n \times h \times w \times c}$ and the target video $X \in \mathbb{R}^{n \times h \times w \times c}$ into latents $\hat{c} \in \mathbb{R}^{n' \times h' \times w' \times d}$ and $\mathbf{x} \in \mathbb{R}^{n' \times h' \times w' \times d}$ by VAE. Here $n$ and $h \times w$ are original temporal/spatial sizes; $n'$, $h'$, $w'$ are latent sizes. With $n_t$ text tokens $t_{\hat{c}}, t_x \in \mathbb{R}^{n_t \times d}$, a naive baseline is to finetune the DiT on the concatenated sequence $[t_{\hat{c}}, \hat{c}, t_x, \mathbf{x}]$[3], following in-context structure-controlled generation (Ju et al., 2025). This often leads to catastrophic forgetting with limited data (Fig. 5 (b), Tab. 2), because (1) DiTs are pre-trained only for generation, not in-context conditioning, and (2) our reference/target pairs lack pixel-aligned priors, making semantic in-context generation much harder. To stabilize training, we adopt Mixture-of-Transformers (MoT) (Liang et al., 2024): a frozen Video Diffusion Transformer (DiT) plus a trainable parallel expert transformer initialized from the backbone. The expert consumes $[t_{\hat{c}}, \hat{c}]$, while the frozen DiT processes $[t_x, \mathbf{x}]$ (see Fig. 4). Each keeps its own query, key, value projections, feed-forward layers, and norms; at each layer, we concatenate Q/K/V and run full attention for two-way information fusion. This shapes references into prompts conditioned on the current generation and routes guidance into the frozen DiT. With MoT, we preserve the backbone's generation ability, boost the training stability, and achieve plug-and-play in-context control independent of DiT architecture (see Tab. 2).

### 3.4 Temporally Biased Rotary Position Embedding

Similar to observations on Rotary Position Embedding (RoPE) (Su et al., 2024) in in-context image generation (Tan et al., 2025), we find that sharing position embedding between the reference condition and the target video is suboptimal: it imposes a false pixel-level spatiotemporal mapping prior, making the model assume a nonexistent mapping between the reference and the target videos, and perform unsatisfactorily (see artifacts in Fig. 5 (c)). Accordingly, we shift the reference prompt's temporal indices by a fixed offset $\Delta$, placing them before all noisy video tokens while keeping spatial indices unchanged (see right bottom of Fig. 4). This removes the spurious prior, matches the temporal order expected by in-context generation, and leads to improved performance (see Tab. 2).

---

[2]The first frame of the reference video is also injected for inheriting the Image-to-Video backbone ability.

[3]Without loss of generality, we assume text and video are jointly modeled with full attention.

Table 1: **Qualitative Comparison.** We compare against the SOTA structure-controlled generation method VACE (Jiang et al., 2025), the base video DiT model CogVideoX-I2V (Yang et al., 2024), the condition-specific variant CogVideoX-I2V (LoRA) (Hu et al., 2022), and the closed-source commercial models Kling/Vidu (Kling, 2025; Vidu, 2025). Overall, *VAP* delivers performance comparable to the closed-source models and, on average, surpasses the other open-source baselines, as a unified and generalizable model. **Red** stands for the best, **Blue** stands for the second best.

| Metrics | Text | Overall Quality | | | Semantic | User Study |
|---|---|---|---|---|---|---|
| Model | Clip Score↑ | Motion Smoothness↑ | Dynamic Degree↑ | Aesthetic Quality↑ | Alignment Score↑ | Preference Rate (%)↑* |
| **Structure-Controlled Methods** | | | | | | |
| VACE (Original) | 5.88 | 97.60 | 68.75 | 53.90 | 35.38 | 0.6% |
| VACE (Depth) | 22.64 | 97.65 | 75.00 | 56.03 | 43.35 | 0.7% |
| VACE (Optical Flow) | 22.65 | 97.56 | **79.17** | 57.34 | 46.71 | 1.8% |
| **DiT Backbone and Condition-Specific Methods** | | | | | | |
| CogVideoX-I2V | 22.82 | **98.48** | 72.92 | 56.75 | 26.04 | 6.9% |
| CogVideoX-I2V (LoRA)† | 23.59 | 98.34 | 70.83 | 54.23 | 68.60 | 13.1% |
| Kling / Vidu‡ | **24.05** | 98.12 | **79.17** | **59.16** | **74.02** | **38.2%** |
| **Ours** | | | | | | |
| Video-As-Prompt (*VAP*) | **24.13** | **98.59** | **77.08** | **57.71** | **70.44** | **38.7%** |

† We fine-tune LoRA on CogVideoX-I2V for each semantic condition in the benchmark and report the average metric as performance.
‡ Kling and Vidu provide dedicated interfaces for each semantic condition; thus, we treat them as condition-specific.
* We report the *preference rate* by aggregating wins over all comparisons. Each cell is the average rate of human preferences received by the corresponding method.

## 4 EXPERIMENTS

### 4.1 IMPLEMENTATION DETAILS

We train *VAP* on CogVideoX-I2V-5B (Yang et al., 2024) and Wan2.1-I2V-14B (Wan et al., 2025) to evaluate effectiveness across DiT architectures.[4] For fairness, we match parameter counts: on CogVideoX-I2V-5B, the in-context DiT expert is a full copy of the original; on Wan2.1-I2V-14B, it is a distributed copy spanning $\frac{1}{4}$ of layers; both are about 5B parameters. Following pre-trained DiTs, we resize videos to $480 \times 720$ (832) and sample 49 frames at 16 fps. We use AdamW with learning rate $1 \times 10^{-5}$ and train for ~20k steps on 48 NVIDIA A100s (Details are in Sec. C.1).

### 4.2 METRICS

We evaluate 5 metrics across three aspects: text alignment, video quality, and semantic alignment. Following prior work (Liu et al., 2025; Jiang et al., 2025), we measure text alignment with CLIP similarity (Radford et al., 2021) and assess video quality using motion smoothness (Radford et al., 2021), dynamic degree (Teed & Deng, 2020), and aesthetic quality (Schuhmann et al., 2022). We also introduce a semantic-alignment score that measures consistency between the reference and generated videos; we submit each video pair and detailed evaluation rules to Gemini-2.5-pro (Comanici et al., 2025) for automatic scoring (Details are in Sec. C.2).

### 4.3 DATASET

Semantic-controlled video generation requires paired reference and target videos sharing the same non-pixel-aligned semantic controls (*e.g.*, concept, style, motion, camera). Prior work mostly relies on a few manually collected videos tailored to specific semantic conditions (Liu et al., 2025), limiting the emergence of unified models. To address this, we collect 2K high-quality reference images from Pexels (Pexels, 2025) spanning men, women, children, animals, objects, landscapes, and multi-subject cases. We then use Image-to-Video visual-effects templates from commercial models (VIDU (Vidu, 2025) and Kling (Kling, 2025)) and community LoRAs (Civitai, 2025) to create paired videos by matching each image to all compatible templates (some restrict subject categories). Overall, we obtain *VAP-Data*, a semantic-controlled paired dataset with over $100K$ samples across 100 semantic conditions—the largest resource (see Sec. 3.2 and Fig. 3). For evaluation, we evenly sampled 24 semantic conditions from 4 categories (concept, style, motion, camera) in the test subset, with 2 samples each. Detailed information and limitations are in Sec. D.

---

[4]As Wan2.1 is more resource-intensive, results are reported on CogVideoX unless otherwise noted.

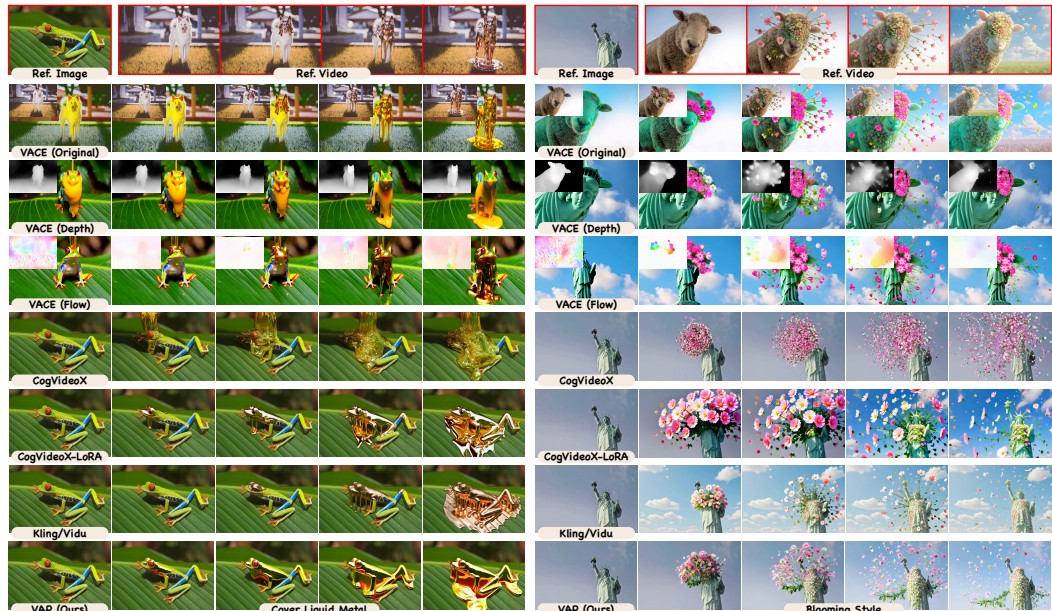

Figure 6: **Qualitative comparison** with VACE (Jiang et al., 2025), CogVideoX (I2V) (Yang et al., 2024), CogVideoX-LoRA (I2V) and commercial models (Kling, 2025; Vidu, 2025); VACE(*) uses a *-form condition (top left). **More visualizations are in Sec. A and the supplementary.**

## 4.4 COMPARISON WITH PREVIOUS METHODS

We evaluate *VAP* against: (1) the state-of-the-art (SOTA) structure-controlled video generation method VACE (Jiang et al., 2025) under multiple structure conditions (*e.g.*, original reference video, depth, optical flow); (2) condition-specific methods, where we train a LoRA (Hu et al., 2022) for each semantic condition—a common community practice often reported to match or surpass task-specific models (Bai et al., 2025a; Ye et al., 2025)—and report averaged performance; (3) state-of-the-art closed-source commercial models, including Kling (Kling, 2025) and Vidu (Vidu, 2025).

**Quantitative Comparison.** For the SOTA structure-controlled method VACE (Jiang et al., 2025), the model conditions on a video and a same-size mask indicating edit (1) vs. fixed (0) regions. Following VACE, we use the reference video, its depth, and its optical flow as video conditions, setting the mask to 1 so the model follows rather than copies them. Overall, VACE performs worst, as expected when structure-controlled methods are applied directly to semantic-controlled generation. This is because VACE assumes a pixel-wise mapping between the condition and the output (*e.g.*, a video and its depth), which breaks under semantic control and copies unwanted appearance or layout from the reference. As control moves from raw video, depth to optical flow, appearance detail decreases, and metrics improve, confirming that the pixel-wise prior is ill-suited for semantic-controlled generation. Driving a pre-trained DiT (CogVideoX-I2V) with captions carrying semantic cues yields decent video quality but weak semantic alignment, since many semantics are hard to express with coarse text. Common LoRA fine-tuning often obtains strong semantic alignment by overfitting a specific condition: it harms base quality (vs. the CogVideoX-I2V row), needs a separate model per condition, and fails to generalize to unseen references. By contrast, *VAP* outperforms open-source baselines on most metrics, achieves performance comparable to commercial models, and, for the first time, provides a unified model for semantic-controlled video generation.

**User Study** We conducted a user study with 20 randomly selected video-generation researchers to evaluate video quality and semantic alignment. In each trial, raters compared different method outputs shown with a semantic-control reference video and chose the better result for (i) semantic alignment and (ii) overall quality. We report the *preference rate*—the normalized share of selections across all comparisons, totaling $100\%$—in Tab. 1. *VAP* and Kling/Vidu (commercial, closed-source, task-specific) achieve the overall highest preference rate, while *VAP* works as a unified model.

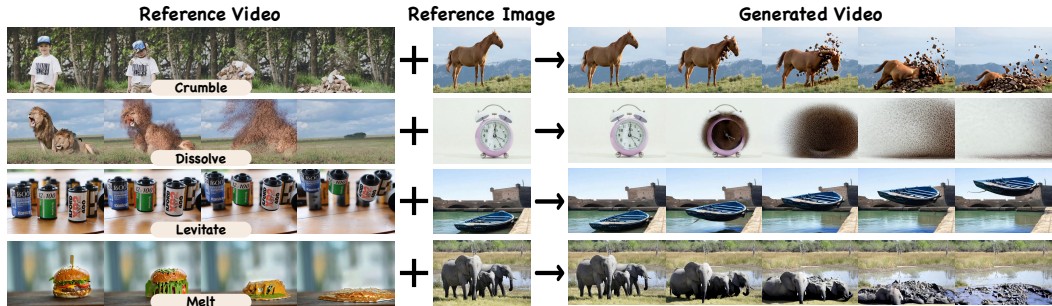

Figure 7: **Zero-Shot Performance.** Given semantic conditions unseen in *VAP-Data* (left column), *VAP* still transfers the abstract semantic pattern to the reference image in a zero-shot manner.

**Qualitative Comparison** In Fig. 6, *VAP* yields better temporal coherence, visual quality, and semantic consistency than structure-controlled baselines (Jiang et al., 2025), DiT backbones, and condition-specific finetuning (Hu et al., 2022), and matches condition-specific commercial models Kling (Kling, 2025) and Vidu (Vidu, 2025). VACE's pixel-mapping bias treats the semantic reference video as pixel-aligned, causing appearance/layout copying (*e.g.*, the frog stands like the dog; the Statue of Liberty imitates a sheep); this artifact weakens when the reference is replaced by depth and then optical flow, which progressively remove appearance details. LoRA fine-tuning improves semantic alignment without copy artifacts but requires a separate model per condition and lacks zero-shot generalization. In contrast, *VAP* uses a single model that treats all semantic conditions as a unified reference-video prompt, enabling unified semantic-controlled generation.

**Zero-Shot Generation** By treating all semantic conditions as unified video prompts, *VAP* supports diverse semantic-controlled generation tasks; moreover, when given an unseen semantic reference (Liu et al., 2025) that doesn't belong to *VAP-Data* (see Fig. 7), the in-context ability learned from video-as-prompt data enables *VAP* to perform zero-shot generation guided by new references.

### 4.5 ABLATION STUDY

**In-Context Generation Structure.** We train 4 *VAP* variants to test the effectiveness of our mixture-of-transformers (MoTs) adoption: **A1. Single-Branch Finetuning** $u_{\Theta}^{s}$: expand pre-trained DiT input sequence to $[Ref_{text}, Ref_{video}, Tar_{text}, Tar_{video}]$ and finetune the full model; **A2. Single-Branch LoRA Finetuning** $u_{\Theta}^{sl}$: same as A1 but freeze the backbone and train only the LoRA layers; **A3. Unidirectional Cross-Attn** $u_{\Theta}^{uc}$: freeze the pre-trained DiT, add a new branch with the same weights, and inject its features via layer-wise cross-attention; and **A4. Unidirectional Addition** $u_{\Theta}^{ua}$: same as A3 but inject features via residual addition. We evaluate on the same benchmark of *VAP-Data*. Results in Tab. 2 show: **A1.** MoT boosts performance by preserving the base DiT's generative ability, solving the catastrophic forgetting while enabling plug-and-play in-context control. **A2.** LoRA helps retain the backbone's ability, but its limited capacity struggles with complex in-context generation, yielding suboptimal results. **A3.** Layer-wise bidirectional information exchange in MoT lets the reference video-prompt representation adapt synchronously to the target tokens, improving semantic alignment. **A4.** Even with retraining, residual-addition methods rely on rigid pixel-to-pixel mapping, mismatching semantic-controlled generation and degrading performance.

**Position Embedding Designs.** To validate the effectiveness of our temporally-biased RoPE, we evaluate two variants. (1) $u_{\Theta}^{i}$: applying identical RoPE to both the reference and target videos, which enforces an unrealistic pixel-wise alignment prior and leads to degraded performance; (2)$u_{\Theta}^{n}$: in addition to introducing a temporal bias $\Delta$, following in-context image generation (Tan et al., 2025), we add a width bias by placing the reference video to the left of the target video. Experiments show that this increases the difficulty of spatial referencing and results in performance degradation.

**Scalability.** As shown in the scalability section, *VAP* improves across all metrics as training data grows, demonstrating strong scalability. This follows from our unified design that treats reference videos as prompts without task-specific modifications, together with the MoT framework, which preserves the backbone's generative capacity while enabling plug-and-play in-context generation.

Table 2: **Ablation Study.** We ablate on the in-context generation structure designs, temporal-biased RoPE, the scalability, and the transferability across different DiT structures. The last row is our default model (VAP), which uses MoT structure, temporal-biased RoPE, 100K training pairs, and CogVideoX-I2V-5B. **Red** stands for the best, **Blue** stands for the second best.

| Metrics Variant | Text CLIP Score ↑ | Overall Quality | | | Semantic Alignment Score ↑ |
| --- | --- | --- | --- | --- | --- |
| | | Motion Smoothness ↑ | Dynamic Degree ↑ | Aesthetic Quality ↑ | |
| **In-Context Generation Structure** | | | | | |
| $u_\Theta^{\text{s}}$ (Unidir-Cross-Attn) | 23.03 | 97.97 | 70.83 | 56.93 | 68.74 |
| $u_\Theta^{\text{sl}}$ (Single-Branch-LoRA) | 23.12 | 98.25 | 72.92 | 57.19 | 69.08 |
| $u_\Theta^{\text{uc}}$ (Unidir-Cross-Attn) | 22.96 | 97.94 | 66.67 | 56.88 | 67.16 |
| $u_\Theta^{\text{ua}}$ (Unidir-Addition) | 22.37 | 97.63 | 62.50 | 56.91 | 55.99 |
| **Position Embedding Design** | | | | | |
| $u_\Theta^{\text{i}}$ (Identical PE) | 23.17 | 98.49 | 70.83 | 57.09 | 68.98 |
| $u_\Theta^{\text{n}}$ (Neg. shift in $T, W$) | 23.45 | **98.53** | 72.92 | 57.31 | 69.05 |
| **Scalability‡** | | | | | |
| $u_\Theta$ (1K) | 22.84 | 92.12 | 60.42 | 56.77 | 63.91 |
| $u_\Theta$ (10K) | 22.87 | 94.89 | 64.58 | 56.79 | 66.28 |
| $u_\Theta$ (50K) | 23.29 | 96.72 | 70.83 | 56.82 | 68.23 |
| **DiT Structure** | | | | | |
| $u_\Theta^{\text{Wan}}$ (Wan2.1-I2V-14B) | **23.93** | 97.87 | **79.17** | **58.09** | **70.23** |
| **Ours** | | | | | |
| $u_\Theta$ (VAP) | **24.13** | **98.59** | **77.08** | **57.71** | **70.44** |

† **Notation.** $u_\Theta$ (our *VAP* parameterized by $\Theta$). s (in-context single-branch finetuning), sl (in-context single-branch LoRA finetuning), uc (unidirectional cross-attention injection), ua (unidirectional residual addition), i (identical position embedding in reference and target), n (temporal shift + negative temporal/width shifts of position embedding), Wan (Wan2.1 as DiT backbone).

‡ **Scale.** $u_\Theta(M)$ indicates the number of training pairs ($M \in \{1\text{K}, 10\text{K}, 50\text{K}, 100\text{K}\}$). Our final version uses 100K training pairs.

**DiT Structure.** To test transferability, we equip Wan2.1-I2V-14B with *VAP* equal in parameter counts to CogVideoX-I2V-5B version (evenly inserted across $\frac{1}{4}$ layers; $\approx 5B$), which—benefiting from Wan2.1's stronger base—improves dynamic degree and aesthetic score but, because the only $\frac{1}{4}$ in-context interaction, yields slightly worse reference alignment than *VAP* on CogVideoX.

We also ablate the in-context expert transformer layer distribution of *VAP*, and the video-prompt representation. Further experiment details are in Sec. F due to page limits.

## 5 CONCLUSION

Video-As-Prompt (*VAP*) is a unified, semantic-controlled video generation framework that treats reference videos as prompts and enables plug-and-play in-context control via a mixture-of-transformers expert. *VAP* overcomes limits of structure-controlled methods (e.g., inappropriate pixel-wise priors) and task/condition-specific designs (e.g., non-generalizable models), providing scalable semantic control and zero-shot generalizability. We build *VAP-Data*, the largest semantic-controlled video generation dataset, and show in extensive experiments that *VAP* achieves state-of-the-art among open-source models, comparable performance to commercial models, and strong generalization.

**Limitations and Future Works.** Despite strong performance, some limitations need further study: (1) We experimented on our large-scale *VAP-Data*, yet the semantic conditions in *VAP-Data* are relatively limited, synthetic, and derived from other generative models, which may inherit the specific stylistic biases, artifacts, and conceptual limitations of the source templates (see Sec. D). We leave the construction of larger-scale, real, semantic-controlled video data to future work. (2) *VAP* uses a reference video, a reference caption, and a target caption to guide semantic control. To stay close to the original DiT distribution, we employ standard video descriptions as captions; however, inaccurate semantics descriptions or large subject mismatch can degrade generation quality (see Sec. E). Instruction-style captions (e.g., "please follow the Ghibli style in the reference video") may more effectively capture the intended semantics and improve control.

ACKNOWLEDGMENTS

This project was supported in part by the Innovation and Technology Fund (UD-1a-MHP/213/24), Hong Kong S.A.R.

## 6  ETHICS STATEMENT

**Scope and intended use (research-only).**  *VAP* targets *semantic-controlled* video generation for research, education, and creative prototyping, where a *reference video* and an optional caption steer concept/style/motion/camera. It is *not* intended for surveillance, impersonation, political persuasion, or other high-risk deployments. We will accompany any artifact release with a research-only license and an acceptable-use policy (AUP) that explicitly prohibits abusive or unlawful scenarios.

**Misuse risks and technical/operational mitigations.**  Potential misuses include identity impersonation, "deepfake" content, targeted harassment, deceptive political messaging, and generation of sexualized or violent media. Our mitigations include: (i) a research-only release; (ii) default content filters blocking clearly harmful categories (e.g., sexual content, explicit violence, hate symbols).

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

In the appendix, we provide more qualitative results (Sec. A), downstream application demonstration (Sec. B), more implementation details (Sec. C), including the hyperparameters and the semantic alignment score metric. Then, we illustrate more dataset details and limitations of our *VAP-Data* (Sec. D). Furthermore, we discuss more about the influence of reference video quality, caption quality, and multiple reference videos (Sec. E). And we conduct more ablation about *VAP* (Sec. F). Finally, we illustrate the use of large language models (Sec. G).

## A  GALLERY

To further demonstrate our *VAP*'s performance, we provide more semantic-controlled generation cases in Fig. 8, Fig. 9. **We strongly encourage readers to view our webpage in the supplementary for better visualization.**

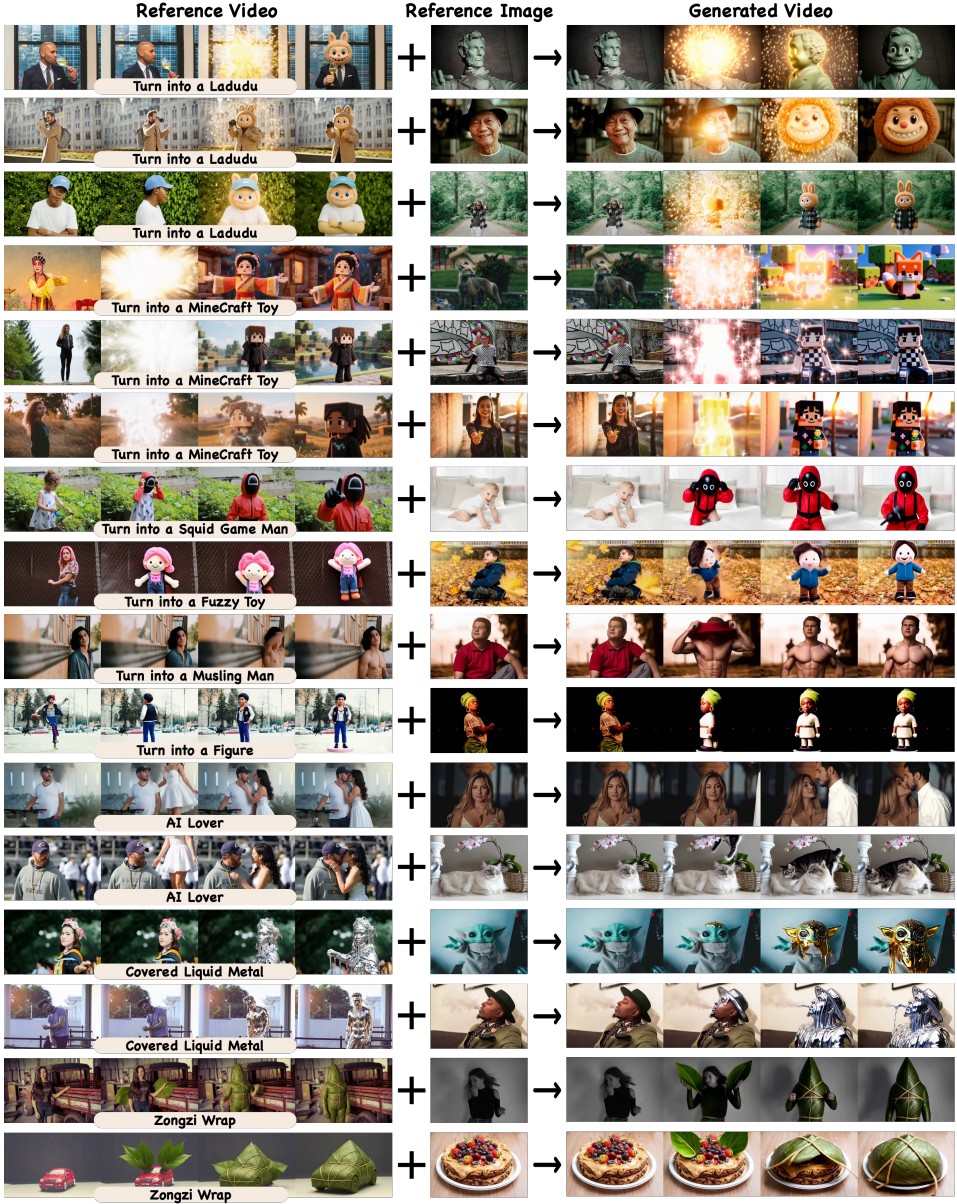

Figure 8: **Additional visualizations** of *VAP*, including entity transformation and entity interaction in concept semantic categories.

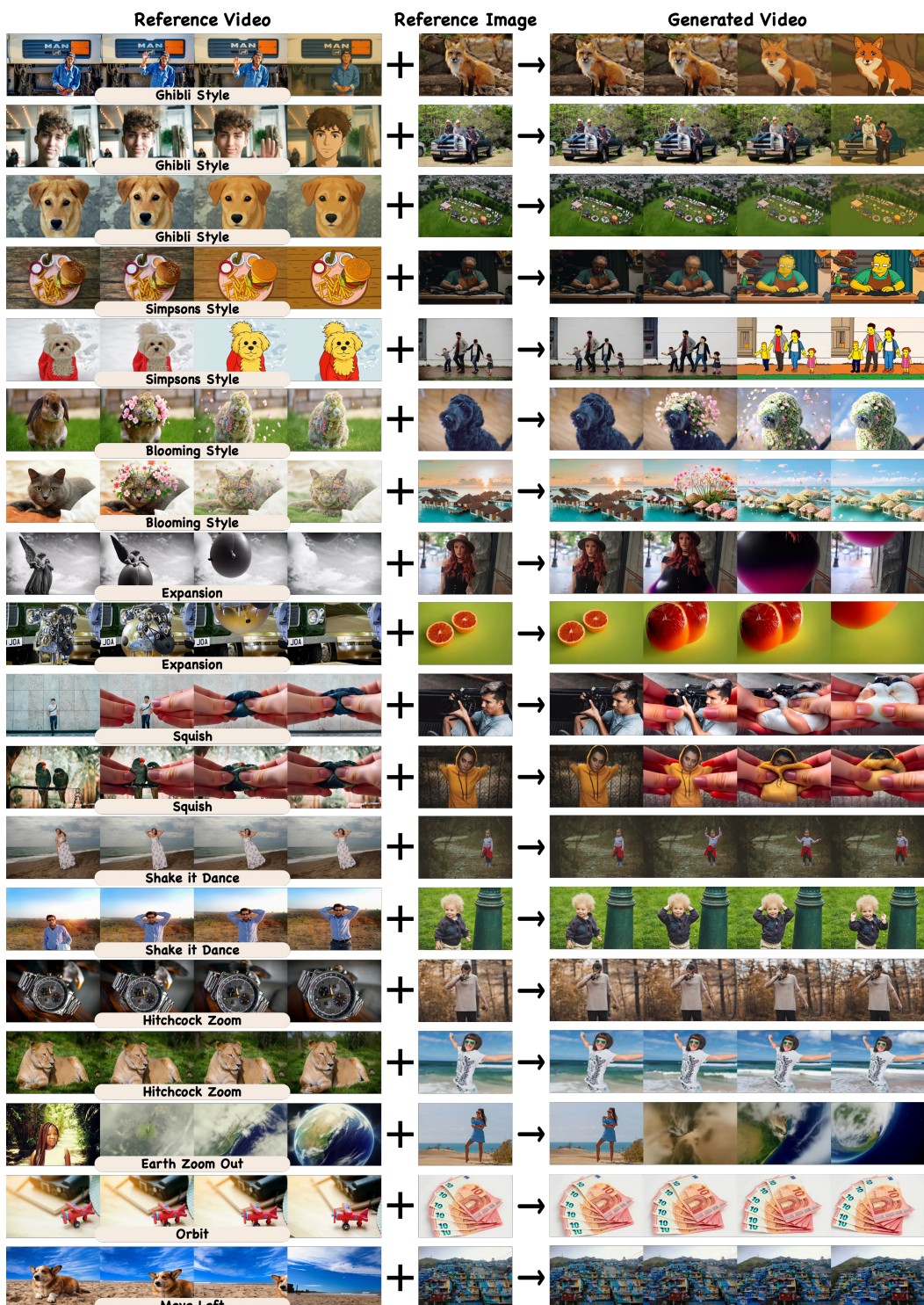

Figure 9: **Additional visualizations** of *VAP*, including style semantic categories, motion semantic categories (Non-Human Motion and Human Motion), and camera semantic categories.

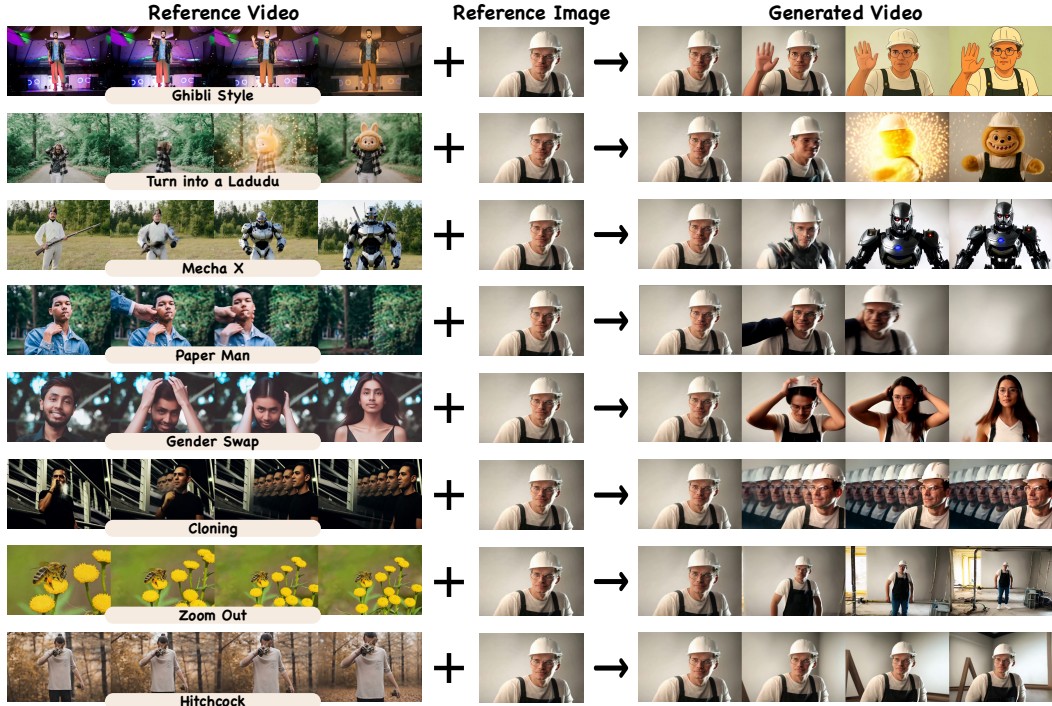

Figure 10: Given different reference videos (with different semantics) and the same reference image, our *VAP* consistently generates a new video for each semantic.

## B APPLICATION

Our Video-as-Prompt (*VAP*) model supports the following downstream applications by disentangling a semantic concept from a source video and applying it to a new subject:

1. Given different reference videos (with different semantics) and the same reference image, our *VAP* consistently generates a new video for each semantic (Fig. 10);

2. Given different reference videos (with the same semantics) and the same reference image, our *VAP* consistently generates the target video aligned with the provided semantics (Fig. 11);

3. Given one reference video and different reference images, our *VAP* transfers the same semantics from the reference video to each image and generates the corresponding videos (Fig. 12);

4. Beyond video prompts, *VAP* allows for fine-grained adjustments using modified text prompts, by fixing the reference inputs and only changing a single word in the prompt (*e.g.*, black to white). *VAP* can precisely edit attributes of the generated output while preserving identity and motion (Fig. 13).

## C IMPLEMENTATION DETAILS

### C.1 HYPERPARAMETERS

In Tab. 3, we summarize hyperparameters for two *VAP* variants based on CogVideoX-5B (Yang et al., 2024) and Wan2.1-14B (Wan et al., 2025), respectively, showing transferability across different DiT architectures.

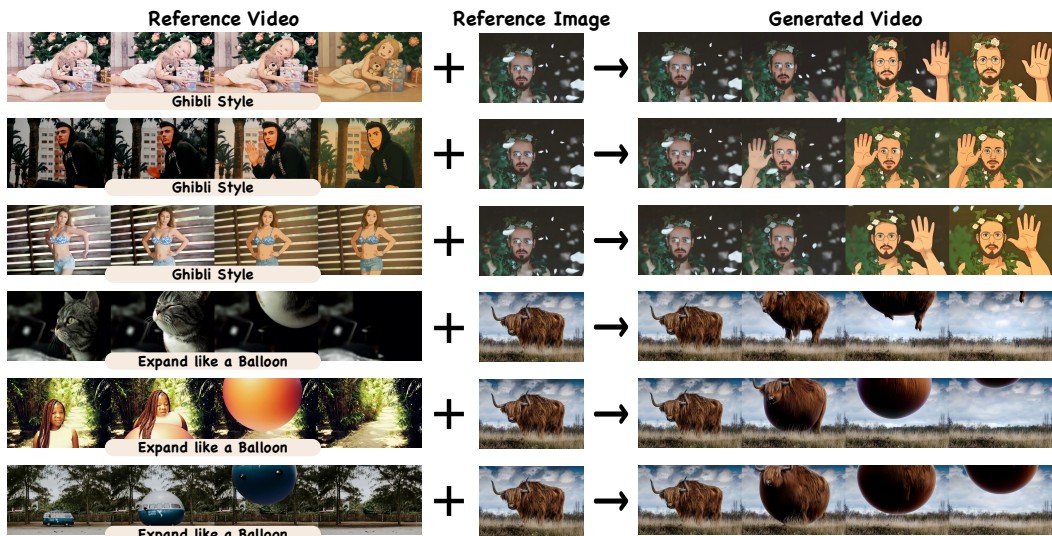

Figure 11: Given different reference videos (with the same semantics) and the same reference image, our *VAP* consistently generates the target video aligned with the provided semantics

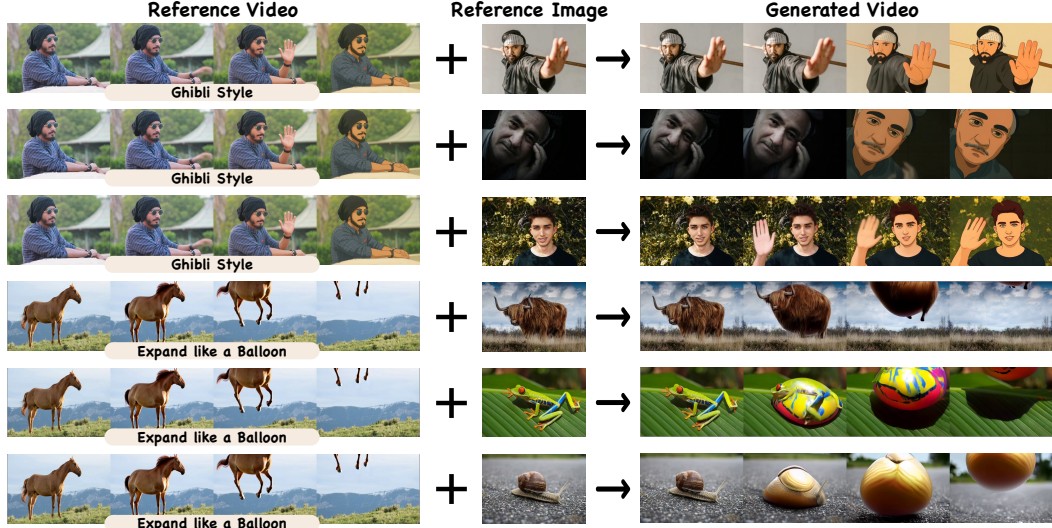

Figure 12: Given one reference video and different reference images, our *VAP* transfers the same semantics from the reference video to each image and generates the corresponding videos.

## C.2 METRICS

As stated in Sec. 4.2, standard video-quality metrics (e.g., CLIP score (Radford et al., 2021), aesthetic score (Schuhmann et al., 2022)) do not reliably capture adherence to a specific semantic condition, so we introduce a semantic-alignment score that measures consistency between the reference semantic condition and the generated video; we submit each (reference, generation) pair and the evaluation rules to Gemini-2.5-pro (Comanici et al., 2025) for automatic scoring.

The evaluation rules pair a general template with key criteria for each semantic; for each case, we provide the template, the criteria for the current semantic (see Tab. 4), the reference video condition, and the generated video to the VLM, which scores them under these rules.

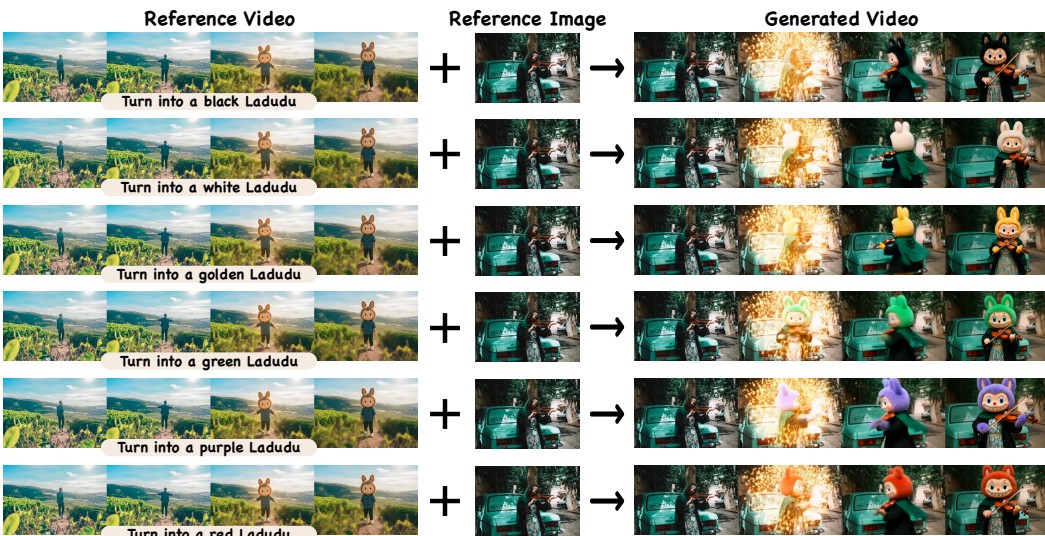

Figure 13: Given a fixed reference video and a reference image, our *VAP* preserves semantics and identity while using a user-modified prompt to adjust fine-grained attributes.

Table 3: Hyperparameter selection for CogVideoX-I2V-5B-based and Wan2.1-I2V-14B-based *VAP*.

| Hyperparameter | Model | |
|---|---|---|
| | CogVideoX-I2V-based | Wan2.1-I2V-based |
| Batch Size / GPU | 1/1 | 1/2 |
| Accumulate Step | 1 | 1 |
| Optimizer | AdamW | AdamW |
| Weight Decay | 0.0001 | 0.0001 |
| Learning Rate | 0.00001 | 0.00001 |
| Learning Rate Schedule | constant with warmup | constant with warmup |
| WarmUp Steps | 1000 | 1000 |
| Training Steps | 20,000 | 20,000 |
| Resolution | 480p | 480p |
| Prediction Type | Velocity | Flow Matching |
| Num Layers | 42 | 40 |
| MoT Layers | [0, 1, 2, 3, 4, 5, 6, 7, 8, 9, 10, 11, 12, 13, 14, 15, 16, 17, 18, 19, 20, 21, 22, 23, 24, 25, 26, 27, 28, 29, 30, 31, 32, 33, 34, 35, 36, 37, 38, 39, 40, 41] | [0, 4, 8, 12, 16, 20, 24, 28, 32, 36] |
| Pre-trained Model | CogVideoX-I2V-5B | Wan2.1-I2V-14B |
| Sampler | CogVideoX DDIM | Flow Euler |
| Sample Steps | 50 | 30 |
| Guide Scale | 6.0 | 5.0 |
| Generation Speed (1 A100) | ~540s | ~420s |
| Device | A100×48 | A100×48 |
| Training Strategy | FSDP / DDP / BFloat16 | FSDP / DDP Parallel / BFloat16 |

To validate the stability of the semantic alignment score, we conduct the same evaluation experiment with another state-of-the-art vision lanuage model GPT-5 (OpenAI, 2025); its scores match closely Gemini-2.5-Pro (Comanici et al., 2025) and follow the trends of human preference rate in our user study (see Tab. 5), confirming the validity of the metric. This further verifies the effectiveness and validity of our proposed semantic alignment score.

Table 4: **Prompt components for the semantic-alignment metrics.** We provide the general template and the specific criteria of "Ghibli Style" as an example.

| Category | Content |
|---|---|
| General Template | You are an expert judge for reference–based semantic video generation. INPUTS REFERENCE video: the target semantic to imitate. TEST video: a new output conditioned on a NEW reference image. Human criteria (treat as ground truth success checklist; overrides defaults if conflict): {criteria} REGIME DECISION Classify the semantics into one of: A) ID-TRANSFORM (identity-changing): the main subject/object changes semantic class or material/state. Layout and identity may legitimately change as a consequence of the transformation. B) NON-ID-TRANSFORM (identity-preserving): stylization, camera motion (pan/zoom), mild geometry exaggeration, lighting changes, human motion, etc. The main subject class/identity should remain the same. If the REFERENCE clearly shows a class/state change, choose A. Otherwise, choose B. When uncertain, choose B. EVALUATION 1) SEMANTIC MATCH (0–60) Regime A (ID-TRANSFORM): How strongly and accurately does TEST reproduce the REFERENCE's target state/look/behavior on the correct regions? Is the source→target mapping consistent (same parts transform to corresponding target parts)? Does the transformed state resemble the REFERENCE target, not a generic filter? Regime B (NON-ID-TRANSFORM): Does TEST replicate the specific semantic (style, camera motion, geometric exaggeration) while keeping the subject recognizable and aligned to the intended scope? 2) IDENTITY / LAYOUT CORRESPONDENCE (0–20) Regime A: Reward semantic correspondence rather than identical identity; coarse scene continuity is preserved unless the REFERENCE implies re-layout. Regime B: Main subject identity stays intact (face/body/clothes/features), and coarse spatial layout remains consistent (no unintended subject swaps/teleports). 3) TEMPORAL QUALITY and TRANSFORMATION CONTINUITY (0–20) Check onset→sustain→offset completeness of the transformation as implied by the REFERENCE. Avoid pop-in/out. Motion is smooth, minimal flicker, and the background is reasonably stable. No frozen loops unless REFERENCE loops. HARD FAIL CAP (force FINAL ¡= 20 if any true) - REFERENCE shows an ID-TRANSFORM, but TEST lacks the transformation, targets the wrong class/material, or completes ¡70% of the transformation timeline. - Severe identity loss in Regime B (unrecognizable face/body, unintended person/object swap). - Gross broken anatomy (detached/missing limbs, implausible face mash) is not required by the semantics. - Extreme temporal instability or unreadable corruption (heavy strobe, tearing, tiling). - Hallucinated intrusive objects that block the subject or derail the semantics. OUTPUT (exactly ONE line of JSON; integer only) {"score": 1–100} |
| Semantic Criteria | Regime: NON-ID-TRANSFORM (identity-preserving stylization). Semantic: Ghibli-style stylization — the overall look gradually transitions to a hand-drawn, soft, film-like Ghibli aesthetic across the whole frame. Identity preservation: The main subject remains recognizable; appearance/proportions/base colors are largely maintained (stylistic simplification and brush-like textures allowed). Motion allowance: Light natural motion is allowed (e.g., slight subject or scene movement) without disrupting effect consistency. Exclusions: No identity swaps, major re-layout, or gross anatomy distortions unless explicitly implied by the reference. . . . |

Table 5: **Semantic alignment score metric and user preference.** Columns are models; rows are semantic alignment score evaluated by Gemini-2.5-Pro (Comanici et al., 2025), semantic alignment score evaluated by GPT-5 (OpenAI, 2025), and human preference rate results of our user study.

| Metric | VACE (Original) | VACE (Depth) | VACE (Optical Flow) | CogVideoX-I2V | CogVideoX-I2V (LoRA) | Kling / Vidu | Video-As-Prompt (*VAP*) |
|---|---|---|---|---|---|---|---|
| Alignment Score (Gemini-2.5-Pro)↑ | 35.38 | 43.35 | 46.71 | 26.04 | 68.60 | 74.02 | 70.44 |
| Alignment Score (GPT-5)↑ | 32.52 | 39.41 | 45.09 | 28.36 | 66.93 | 73.91 | 70.26 |
| Preference Rate (%)↑ | 0.6% | 0.7% | 1.8% | 6.9% | 13.1% | 38.2% | 38.7% |

# D  DATASET

## D.1  DATASET DETAILS

In-context learning requires vast amounts of example pairs, which simply do not exist for semantic video tasks. Filming $100k$ real-world pairs is nearly impossible for a research exploration. Our solution was to bootstrap it. We curated thousands of high-quality real images and then used the existing "zoo of specialist models" (commercial APIs (Kling, 2025; Vidu, 2025; PixVerse, 2025) and LoRAs (Hu et al., 2022; Civitai, 2025)) as a powerful, automated engine to create our paired dataset, *VAP-Data*. As shown in Sec. 3.2 and Fig. 3, *VAP-Data* is the largest semantic-controlled paired dataset to date, with over $100K$ samples across 100 semantic conditions, covering 4 primary categories: concept (entity transformation and interaction), style, motion (human and non-human), and camera movement. The detailed distribution of semantic conditions is provided in Tab. 6.

Crucially, *VAP-Data* is more than just a dataset; it's proof of a concept. We show that we can train a single generalist model (*VAP*) to learn the unified underlying principle of semantic control by showing it various examples from disparate specialist models.

For evaluation, we evenly sampled 24 semantic conditions from 4 categories (concept, style, motion, camera) in *VAP-Data* test subset, with 2 samples each, totaling 48 test samples.

## D.2  DATASET LIMITATIONS

Even though our *VAP-Data* is the largest semantic-controlled video generation dataset, it still has limitations. As noted in Sec. 4.3, *VAP-Data* was created using visual effects templates from commercial models (vidu (Vidu, 2025), Kling (Kling, 2025)) and community LoRAs (Hu et al., 2022; Wan et al., 2025; Yang et al., 2024; Civitai, 2025). Thus, the dataset is synthetic and derived from other generative models, leading to *VAP* may inherit the specific stylistic biases, artifacts, and conceptual limitations of the source templates (e.g., if the source models are poor at generating hands, *VAP* will likely not learn to generate hands well from this data). Building a large, real-world, semantic-controlled video dataset would help address this issue, but it is beyond this paper's main focus; we leave it for future work.

Nevertheless, zero-shot experiments in Sec. 4.4 and downstream tasks in Sec. B show that *VAP* generalizes to unseen semantic conditions (Liu et al., 2025) (e.g., crumble, dissolve, levitate, melt) and across tasks, including using different reference videos to prompt a single reference image under different semantic conditions or using the same reference videos to prompt different reference images under a fixed semantic condition. These results demonstrate the generality of *VAP* and we hope they inspire advances in controllable video generation; broader data collection is left to future work. Additional visualizations are available on the supplementary webpage.

# E  LIMITATION ANALYSIS

## E.1  INFLUENCE OF REFERENCE VIDEO AND CAPTION

*VAP* learns in-context generation from large paired video–caption data: given captions for a reference and a target video, the shared semantic attributes in both captions aid in transferring the semantic properties of the reference video to the target video. Specifically, when both captions mention the same concept (*e.g.*, "molten metal pours over the target . . . ") in a similar way, *VAP* retrieves the relevant semantics from the reference prompt and applies it to the target. The reason why we use standard video-description captions (*e.g.*, ". . . A static Grogu is centered. . . A viscous, reflective

Table 6: **Dataset statistics by** 4 **primary semantic categories.** We reorganize the dataset into 4 primary categories: *Concept* (merging entity transformation and interaction), *Style*, *Motion* (covering human and non-human motion transfer), and *Camera Movement*. For each primary category, we report its subcategory (if any), the alphabetical semantic condition subset list (names come from commercial models API definition (Kling, 2025; Vidu, 2025), and community visual effects LoRA definition (Civitai, 2025), see Sec. 4.3), and the total number of videos.

| Primary Category | Subcategory | Subset (alphabetical) | Total Videos |
|---|---|---|---|
| **Concept** (n=56) | | | |
| | Entity Transformation (n=24) | captain america, cartoon doll, eat mushrooms, fairy me, fishermen, fuzzyfuzzy, gender swap, get thinner, hair color change, hair swap, laudu me, mecha x, minecraft, monalisa, muscling, pet to human, sexy me, squid game, style me, super saiyan, toy me, venom, vip, zen | 17k |
| | Entity Interaction (n=21) | aliens coming, child memory, christmas, cloning, couple arrival, couple drop, couple walk, covered liquid metal, drive yacht, emoji figure, gun shooting, jump to pool, love drop, nap me, punch hit, selfie with younger self, slice therapy, soul depart, watermelon hit, zongzi drop, zongzi wrap | 20k |
| **Style** (n=11) | | | |
| | Stylization (n=11) | american comic, bjd, bloom magic, bloombloom, clayshot, ghibli, irasutoya, jojo, painting, sakura season, simpsons comic | 15k |
| **Motion** (n=41) | | | |
| | Human Motion Transfer (n=16) | break glass, crying, cute bangs, emotionlab, flying, hip twist, laughing, live memory, live photo, pet belly dance, pet finger, shake it dance, shake it down, split stance human, split stance pet, walk forward | 10k |
| | Non-human Motion Transfer (n=16) | auto spin, balloon flyaway, crush, decapitate, dizzydizzy, expansion, explode, grow wings, head to balloon, paperman, paper fall, petal scattered, pinch, rotate, spin360, squish | 19k |
| **Camera** (n=12) | | | |
| | Camera Movement Control (n=12) | dolly effect, earth zoom out, hitchcock zoom, move down, move left, move right, move up, orbit, orbit dolly, orbit dolly fast, zoom in, zoom out | 19k |

‡ Subset counts (n) are reported per subcategory and are alphabetically sorted within each subcategory.
**Overall subsets across all primary categories: 100. Overall videos across all categories:** $> 100$**k.**

gold liquid appears on the forehead ...", "A young woman stands still... A thick, reflective liquid metal begins to pour over her face from above..."), is to match the pre-training data distribution, Consequently, performance depends on caption quality and on structural similarity between the main subjects: it is stable when caption styles align and subjects are similar, but degrades when descriptions diverge (*e.g.*, "...A viscous, reflective **gold liquid** appears on the forehead ..." vs. "...A viscous, reflective rose-gold **water** pours over the snail ...") or when subjects differ markedly (*e.g.*, **Grogu vs. snail**). As shown in Fig. 14, the bad caption mislabels "water" instead of the intended "liquid metal"; the good reference subject (the young woman) is structurally closer to Grogu, while the snail differs greatly and its semantic signal is weak (the liquid metal and shell have similar colors), yielding poorer alignment and less appealing visuals for the bad reference case.

We empirically observe that Video-As-Prompt typically:

1. First, relies on the **most critical shared semantics** in the reference and target captions to establish a **coarse-level "semantic anchor"** for the concept to be transferred.

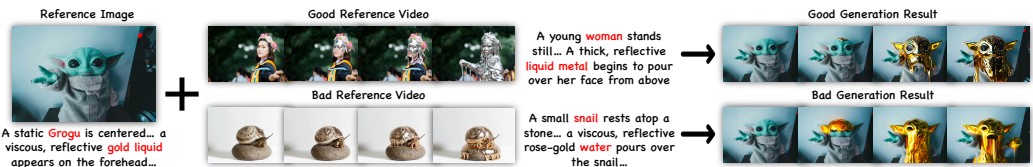

Figure 14: **Limitation visualization.** *VAP* transfers semantics reliably when the semantic description of reference caption aligns with that of the target caption and subject structure aligns with the target: aligned descriptions ("gold liquid" and "liquid metal") and similar subject structures (Grogu and a young woman) yield good results (top). Mislabeled semantic descriptions ("water" vs. "liquid metal"), or large subject mismatch (Grogu vs. snail), reduce alignment and visual quality (bottom).

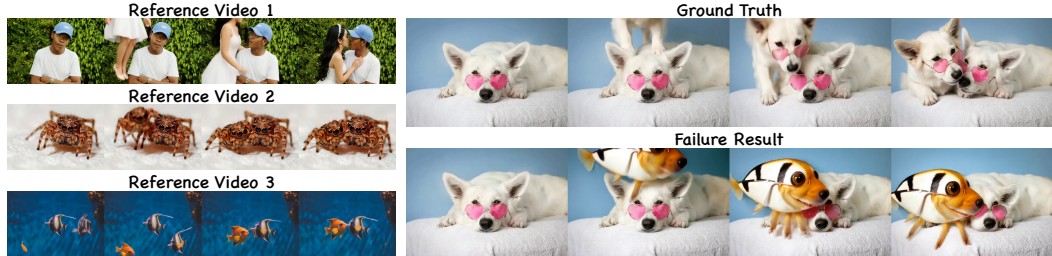

Figure 15: **Failure case of multi-reference prompting.** Left: three reference videos with divergent structure and similar semantics (human, spider, flatfish). Right: Ground truth is on top. Using three (bottom) spuriously transfers unwanted appearance cues (e.g., fish shape and spider-like legs) onto the dog. We attribute this leakage to generic captions that lack an explicit referent; stronger multi-reference control or instruction-style captions could mitigate it.

2. Then, it leverages the **reference video** to fill in the finer-grained details (e.g., stylistic attributes, motion patterns, or camera speed).

When captions are noisy, under-specified, or semantically misaligned, this "anchor" becomes ambiguous. This ambiguity directly **impairs the model's semantic selection** and leads to the observed performance degradation.

### E.2 INFLUENCE OF MULTIPLE REFERENCE VIDEOS

We examine how the number of video prompts affects performance by supplying 1–3 semantically matched reference videos during training and testing. Empirically, results are similar to using a single reference. However, with multiple references, the model may blend unwanted visual details across videos, as shown in Fig. 15. We hypothesize this stems from our general-purpose captions, which lack explicit semantic referents. When the three references differ in structure (human, spider, flatfish) and in semantic realization (e.g., reference 1 clearly shows "AI Lover Drop"; reference 2 introduces a falling spider without a hug; reference 3 is weakest, with a flatfish swimming up instead of falling), the model mixes semantics from reference 1 with appearance from reference 2 (spider legs) and contours from reference 3 (fish shape). A more effective multi-reference control mechanism (*e.g.*, a tailored RoPE for multi-reference conditions)—or an instruction-style caption that specifies the intended referent—may mitigate this issue. A full study of model and caption design for multi-reference training is beyond this work and left for future research.

### E.3 EFFICIENCY

Like prior plug-and-play methods (Zhang et al., 2023; Jiang et al., 2025), our approach avoids retraining pre-trained video diffusion transformers at pre-training scale, but the added parameters introduce extra inference cost—higher memory use and longer runtime. Specifically, the impact varies with the distribution of MoT layers in *VAP*; as shown in Tab. 3, inference time roughly doubles on average, mainly due to additional in-context computation. Given the strong plug-and-play unified semantic control in in-context generation and the fact that we avoid retraining the backbone, this overhead is acceptable. Performance optimizations (e.g., sparse attention (Dao, 2024; Zhang et al.,

2025a) and pruning (Fang et al., 2025; Xie et al., 2025)) are orthogonal and beyond the scope of this work; we leave them to future work.

## F  ABLATION STUDY

Table 7: **Ablation Study.** We verify the effectiveness of our MoT structure, temporal-biased RoPE, the scalability, and the transferability in different DiTs. The bottom row reports our full model.

| Metrics
Variant | Text
CLIP Score ↑ | Overall Quality
Motion Smoothness ↑ | Dynamic Degree ↑ | Aesthetic Quality ↑ | Reference
Alignment Score ↑ |
|---|---|---|---|---|---|
| $u_\Theta^{\text{s}}$  (Single-Branch) | 23.03 | 97.97 | 70.83 | 56.93 | 68.74 |
| $u_\Theta^{\text{sl}}$  (Single-Branch-LoRA) | 23.12 | 98.25 | 72.92 | 57.19 | 69.28 |
| $u_\Theta^{\text{uc}}$  (Unidir-Cross-Attn) | 22.96 | 97.94 | 66.67 | 56.88 | 67.16 |
| $u_\Theta^{\text{ua}}$  (Unidir-Addition) | 22.37 | 97.63 | 62.50 | 56.91 | 55.99 |
| **Position Embedding Design** | | | | | |
| $u_\Theta^{\text{i}}$  (Identical PE) | 23.17 | 98.49 | 70.83 | 57.09 | 68.98 |
| $u_\Theta^{\text{n}}$  (Neg. shift in $T, W$) | 23.45 | 98.53 | 72.92 | 57.31 | 69.05 |
| **Scalability‡** | | | | | |
| $u_\Theta(\text{1K})$ | 22.84 | 92.12 | 60.42 | 56.77 | 63.91 |
| $u_\Theta(\text{10K})$ | 22.87 | 94.89 | 64.58 | 56.79 | 66.28 |
| $u_\Theta(\text{50K})$ | 23.29 | 96.72 | 70.83 | 56.82 | 68.23 |
| $u_\Theta(\text{100K})$ | **24.13** | **98.59** | 77.08 | 57.71 | **70.44** |
| **DiT Structure** | | | | | |
| $u_\Theta^{\text{Wan}}$  (Wan2.1-I2V-14B) | 23.93 | 97.87 | **79.17** | **58.09** | 70.23 |
| **In-Context Expert Transformer Layer Distribution‡** | | | | | |
| $u_\Theta(\mathcal{L}_{\text{odd}})$ | 24.05 | 98.52 | 75.00 | 57.58 | 70.22 |
| $u_\Theta(\mathcal{L}_{\text{odd}, \le \lfloor 0.5 N_l \rfloor})$ | 23.72 | 98.19 | 70.83 | 56.71 | 69.61 |
| $u_\Theta(\mathcal{L}_{\text{first-half}})$ | 23.90 | 98.41 | 75.00 | 57.18 | 69.94 |
| $u_\Theta(\mathcal{L}_{\text{first-last}})$ | 23.96 | 98.33 | 72.92 | 57.06 | 70.02 |
| **Video Prompt Representation** | | | | | |
| $u_\Theta^{\text{n\_ref}}$  (noisy reference) | 23.98 | 98.41 | 75.00 | 57.42 | 70.18 |
| **Ours** | | | | | |
| $u_\Theta$  (VAP) | **24.13** | **98.59** | 77.08 | 57.71 | **70.44** |

† **Notation.** $u_\Theta$ (our VAP parameterized by $\Theta$). s (in-context single-branch finetuning), sl (in-context single-branch LoRA finetuning), uc (unidirectional cross-attention injection), ua (unidirectional residual addition), i (identical position embedding in reference and target), n (temporal shift + negative temporal/width shifts of position embedding), Wan (Wan2.1 as DiT backbone). n_ref (noisy reference prompts).

‡ **MoT layers.** $u_\Theta(\mathcal{L})$ activates MoT blocks on layer index set $\mathcal{L} \subseteq [N_l] = \{1, \ldots, N_l\}$ of the backbone with $N_l$ Transformer layers. We instantiate $\mathcal{L}_{\text{first-half}} = \{1, 2, \ldots, \lfloor 0.5 N_l \rfloor\}$, $\mathcal{L}_{\text{first-last}} = \{1, N_l\}$, $\mathcal{L}_{\text{odd}, \le \lfloor 0.5 N_l \rfloor} = \{1, 3, \ldots, \lfloor 0.5 N_l \rfloor\}$, and $\mathcal{L}_{\text{odd}} = \{1, 3, \ldots, N_l\}$.

§ **Scale.** $u_\Theta(M)$ indicates the number of video training pairs used ($M \in \{1K, 10K, 50K, 100K\}$).

**In-context Generation Structure.** We train $4$ VAP variants to test the effectiveness of our mixture-of-transformers (MoTs) adoption: **A1. Single-Branch Finetuning $u_\Theta^{s}$**: expand pre-trained DiT input sequence to $[Ref_{text}, Ref_{video}, Tar_{text}, Tar_{video}]$ and finetune the full model; **A2. Single-Branch LoRA Finetuning $u_\Theta^{sl}$**: same as A1 but freeze the backbone and train only the LoRA layers; **A3. Unidirectional Cross-Attn $u_\Theta^{uc}$**: freeze the pre-trained DiT, add a new branch with the same weights, and inject its features via layer-wise cross-attention; and **A4. Unidirectional Addition $u_\Theta^{ua}$**: same as A3 but inject features via residual addition. We evaluate on the same benchmark of VAP-Data. Results in Tab. 2 show: **A1.** MoT boosts performance by preserving the base DiT's generative ability while enabling plug-and-play in-context control. **A2.** LoRA helps retain the backbone's ability, but its limited capacity struggles with complex in-context generation, yielding suboptimal results. **A3.** Layer-wise bidirectional information exchange in MoT lets the reference video-prompt representation adapt synchronously to the target tokens, improving semantic alignment. **A4.** Even with new data, residual-addition methods rely on rigid pixel-to-pixel mapping, mismatching semantic-controlled generation and degrading performance.

**Position Embedding Designs.** To validate the effectiveness of our temporally-biased RoPE, we evaluate two variants. (1) $u_\Theta^{i}$: applying identical RoPE to both the reference and target videos, which enforces an unrealistic pixel-wise alignment prior and leads to degraded performance; (2)$u_\Theta^{n}$: in addition to introducing a temporal bias $\Delta$, following in-context image generation (Tan et al., 2025), we add a width bias by placing the reference video to the left of the target video. Experiments show that this increases the difficulty of spatial referencing and results in performance degradation.

**Scalability.** As shown in Tab. 2, *VAP* improves across all metrics as training data grows, demonstrating strong scalability. This follows from our unified design that treats reference videos as prompts without task-specific modifications, together with the MoT framework, which preserves the backbone's generative capacity while enabling plug-and-play in-context generation.

**DiT Structure.** To test transferability, we equip Wan2.1-I2V-14B with *VAP* equal in parameter counts to CogVideoX-I2V-5B version (evenly inserted across $\frac{1}{4}$ layers; $\approx 5B$), which—benefiting from Wan2.1's stronger base—improves dynamic degree and aesthetic score but, because the only $\frac{1}{4}$ in-context interaction, yields slightly worse reference alignment than *VAP* on CogVideoX.

**Mixture-of-Transformers Layer Distribution** We analyze how different layer distributions affect our in-context DiT Expert. $(1)u_\Theta(\mathcal{L}_{\text{first-half}})$: initializing and copying from the first half of the pretrained DiT; $(2)u_\Theta(\mathcal{L}_{\text{first-last}})$: from the first and last layers; $(3)u_\Theta(\mathcal{L}_{\text{odd},\leq\lfloor 0.5N_l\rfloor})$: from the odd layers of the first half; and $(4)u_\Theta(\mathcal{L}_{\text{odd}})$: from all odd layers. The results show that balanced feature interaction improves generation quality ($u_\Theta(\mathcal{L}_{\text{first-last}})$ outperforms $u_\Theta(\mathcal{L}_{\text{first-half}})$, and $u_\Theta(\mathcal{L}_{\text{odd}})$ outperforms $u_\Theta(\mathcal{L}_{\text{odd},\leq\lfloor 0.5N_l\rfloor})$). However, while reducing layers can improve training and inference efficiency, it inevitably harms certain aspects of performance ($u_\Theta(VAP)$ outperforms $u_\Theta(\mathcal{L}_{\text{odd}})$).

**Video Prompt Representation** Inspired by Diffusion Forcing (Chen et al., 2024a; Song et al., 2024; Guo et al., 2025b), we study video prompt representation by injecting noise into it. However, this often leads to severe artifacts. The core reason is that, unlike long-video generation in Diffusion Forcing, where copy-paste or overly static results are common, our reference videos already differ significantly in appearance and layout from the target videos. Thus, adding noise to the video prompt corrupts the contextual information and degrades generation quality.

# G USE OF LARGE LANGUAGE MODELS (LLMS)

**Scope of use.** We used a large language model (LLM) *only for writing polish*, including grammar correction, phrasing refinement, and improvements to clarity and readability. The LLM did *not* contribute to research ideation, problem formulation, method design, experimental setup, result selection, interpretation, or drafting of technical content (theorems, algorithms, proofs, metrics, or analyses). All technical claims, experiments, figures, tables, and conclusions were conceived, implemented, and verified by the authors.

**Process and safeguards.** LLM assistance was applied post hoc to author-written passages to improve presentation quality, without introducing new technical material or references. We reviewed each edited passage for accuracy and faithfulness to the original meaning and ran standard plagiarism checks. No proprietary or sensitive data were disclosed to the LLM beyond non-sensitive manuscript text; when necessary, potentially identifying details were redacted.

**No material impact on research outcomes.** The use of LLMs had no bearing on the research ideas, empirical results, evaluation protocols, or conclusions reported in this paper.

