# OpenReview forum: "Video-As-Prompt: Unified Semantic Control for Video Generation"
_ICLR.cc/2026/Conference — ICLR 2026 Poster_

### Official Review · Reviewer_v4tD · 2025-10-19

**Soundness:** 4
**Presentation:** 4
**Contribution:** 3
**Rating:** 6
**Confidence:** 5

**Summary:**

The paper introduces Video-As-Prompt (VAP), a unified framework for semantic-controlled video generation. Specifically, VAP treats the input video as a semantic prompt, which is used to guide a frozen Video DiT by a plug-and-play Mixture-of-Transformers (MoT) expert. VAP includes three key innovations: 1) in-context MoT for bidirectional information fusion between reference and target videos; 2) a temporally biased RoPE to remove false pixel-wise alignment priors and improve temporal coherence; 3) VAP-Data, a large-scale dataset with over 100K paired videos across 100 semantic conditions (e.g., concept, style, motion, and camera). The experiments show that VAP achieves a 38.7% user preference rate, achieving comparable results with commercial models (e.g., Kling and Vidu), and outperforming existing open-source models.

**Strengths:**

- The most impressive strength of the work is the wide coverage of diverse semantic control tasks, including the control on concept, style, motion, and camera, under a single framework. To the best of my knowledge, VAP is the first model to achieve this level of task unification for IV2V generation which can substantially enhance the community interest.
- While the proposed architecture looks like ControlNet (i.e., a trainable copy alongside of a frozen model), the proper adaption to this task (as shown in Figure 2) sounds novel and makes sense to me. Since the main goal is similar (i.e., ControlNet handle 2D spatial control from reference images while VAP extends to 3D temporal control from reference videos), but the proper adaption could make the model better generalize to different tasks.
- The proposed temporally biased RoPE is also elegant and sounds well-motivated to me as it can avoid 1-to-1 positional mapping between reference and target frames which would otherwise cause false pixel matching and copy-paste artifacts.
- I have checked the generated video samples provided in the supplementary material and they look high-quality. In addition, based on the quantitative results provided in Table 1, the model matches or even exceeds commercial models (e.g., Kling and Vidu) on several metrics, which show a promising and strong performance.
- The paper is well written and easy to follow. The figures are well-plotted and informative which can make readers quickly understand the core ideas.

**Weaknesses:**

- My major concern is the construction of VAP-Data, which is entirely synthesized from the existing models or checkpoints (such as Kling, Vidu and the LoRAs from the community). As a result, VAP may overfit to the semantic controls which have already been supported by the off-the-self models. Under this perspective, the only contribution of the paper becomes an architecture design to unify the existing capabilities instead of the generalization to other novel semantic control. It could be insufficient for an ICLR-level contribution.
- While the authors claim VAP can achieve zero-shot generalization in Figure 7, the demonstrated effects (such as “crumble,” “dissolve,” “levitate,” and “melt”) are already supported by the existing models (e.g., Pika, Kling). It is unclear how truly novel these conditions are for the zero-shot evaluation.
- The real-world evaluation is also limited. To verify the generalization of the model, the authors can consider selecting some famous movie/TV shows/meme clips with interesting semantic control as the references. Such effects could serve as stronger out-of-distribution and real-world references, which can more robustly evaluate the generalizability of the model.
- While the authors provide extensive ablation studies in Table 2, the impact of each proposed component seems very marginal. Could the author provide more theoretical analysis on this? Otherwise, it is unconvincing that the architecture of VAP is promising.
- While I appreciate the authors provide thorough limitation analysis in Appendix E, the performance degradation looks significant when the reference and target captions/objects mismatch. It could significantly limit practical usage, since the consistency of reference and target captions/objects cannot always be guaranteed. Could the authors explain the potential reason and solution for this issue? Could re-captioning videos or randomly combining videos with different objects but same effect mitigate this issue?

**Questions:**

- Could the authors provide more samples using out-of-distribution or real-world videos as the reference semantic control?
- Could the authors explain why the proposed modules only provide marginal improvements over the baseline model?
- Could the authors provide the discussion and potential solution regarding the performance drop when reference and target captions/objects mismatch?
- Since the collection of VAP-Data is claimed as one major contribution, will the dataset be released to the community?

---

> ### Author Response · Authors · 2025-11-19
> **Response to Reviewer v4tD (1/5)**
>
> We express our sincere gratitude to the reviewer for the insightful comments and the recognition of our work, including the soundness of problem reformulation, framework novelty, extrodinary results and presentation. Following the feedback, we have added experiments and addressed the specific concerns raised by the reviewer as detailed below. We believe these enhancements significantly contribute to the clarity and impact of our research.
>
>
>
> > ### Weakness 1: Synthetic Data Limitations
>
> We acknowledge the limitation regarding synthetic data and have discussed this in **Sec. 5 (Conclusion)** and **Appendix D (Dataset)**. The primary reason for choosing synthetic construction is that creating **a large-scale, real-world, semantically controlled video dataset** is *infeasible* at this exploratory stage due to **extremely high annotation costs**.
>
> However, we believe our VAP-Data and the Video-As-Prompt framework offer significant contributions to the community in three key areas:
>
> 1.  **Data Contribution.** Although VAP-Data is synthetic, it is, to the best of our knowledge, the **largest available dataset** for semantic control in video generation. The previous largest dataset, **Open-VFX** ([https://huggingface.co/datasets/sophiaa/Open-VFX](https://huggingface.co/datasets/sophiaa/Open-VFX)), is also synthetic and contains only 559 samples across 14 semantic conditions. In contrast, VAP-Data expands the scale by over **178×, offering more than 100k videos across 100 semantic conditions**. This provides **the first practical, training-friendly data foundation** for *unified* semantic-controlled video generation.
>
> 2.  **Research Contribution.** Our primary goal is to study **how to transfer abstract semantics from a reference video to a target video** under a unified formulation. Our *Video-As-Prompt* approach, combined with the MoT expert and temporally-biased RoPE, validates this core idea. To our knowledge, this is the **first work in the community to validate the "video-as-prompt" concept for unified semantic control**, filling an important gap in controllable video generation.
>
> 3.  **Generalization to Real-World Data.** To further assess generalization, we have **added zero-shot experiments** in the revised manuscript. These demonstrate that although VAP is trained solely on synthetic data, it **generalizes well to real-world reference videos and complex composite semantics**. For example, our model can reproduce:
>     * The complex semantics of a character raising both arms with their back to an explosion (e.g., from the "Iron Man" movie) [anonymous link: [https://anonymous.4open.science/r/Video-As-Prompt-Review-CA61/rebuttal/real\_video\_example\_1.mp4](https://anonymous.4open.science/r/Video-As-Prompt-Review-CA61/rebuttal/real_video_example_1.mp4)].
>     * The famous real-world "Michael Bay shot" (a rapid, low-angle, orbiting shot) [anonymous link: [https://anonymous.4open.science/r/Video-As-Prompt-Review-CA61/rebuttal/real\_video\_example\_2.mp4](https://anonymous.4open.science/r/Video-As-Prompt-Review-CA61/rebuttal/real_video_example_2.mp4)].
>     * Complex composite effects, such as transforming into a robot (i.e., "Transformer") while the environment simultaneously turns to winter [anonymous link: [https://anonymous.4open.science/r/Video-As-Prompt-Review-CA61/rebuttal/multi\_vfx\_example.mp4](https://anonymous.4open.science/r/Video-As-Prompt-Review-CA61/rebuttal/real_video_example_3.mp4)].
>
> Finally, we explicitly acknowledge the importance of constructing a large-scale, real-world, semantic-controlled video dataset. As stated in Line 480, we identify this as a critical direction for future work.

---

> ### Author Response · Authors · 2025-11-19
> **Response to Reviewer v4tD (2/5)**
>
> > ### Weakness 2 & 3, Question 1: More Robust Zero-shot Evaluation
>
> We thank the reviewer for this insightful suggestion. We fully agree on the need for **stronger out-of-distribution and real-world evaluation**. In response, we have added **real-world reference videos** and **more complex composite effect semantics** to comprehensively showcase our zero-shot generalization capabilities.
>
>
> 1.  **Additional Zero-Shot Experiments.** We have added new qualitative results demonstrating generalization to complex real-world scenes (e.g., "Iron Man" pose, "Michael Bay" camera movement, and composite "Transformer + Winter" effects).
>     * The complex semantics of a character raising both arms with their back to an explosion (e.g., from the "Iron Man" movie) [anonymous link: [https://anonymous.4open.science/r/Video-As-Prompt-Review-CA61/rebuttal/real\_video\_example\_1.mp4](https://anonymous.4open.science/r/Video-As-Prompt-Review-CA61/rebuttal/real_video_example_1.mp4)].
>     * The famous real-world "Michael Bay shot" (a rapid, low-angle, orbiting shot) [anonymous link: [https://anonymous.4open.science/r/Video-As-Prompt-Review-CA61/rebuttal/real\_video\_example\_2.mp4](https://anonymous.4open.science/r/Video-As-Prompt-Review-CA61/rebuttal/real_video_example_2.mp4)].
>     * Complex composite effects, such as transforming into a robot (i.e., "Transformer") while the environment simultaneously turns to winter [anonymous link: [https://anonymous.4open.science/r/Video-As-Prompt-Review-CA61/rebuttal/multi\_vfx\_example.mp4](https://anonymous.4open.science/r/Video-As-Prompt-Review-CA61/rebuttal/real_video_example_3.mp4)].
>
> 2.  **Novelty of Current Zero-shot Evaluation.** We acknowledge that specific effects in Figure 7 (e.g., *“crumble,” “dissolve”*) are supported by commercial tools like Pika and Kling. However, we respectfully highlight the importance of our approach as clarified in **Figure 7** and **Lines 192–196 / 410–413**:
>     * The semantic conditions in our zero-shot experiments are **never seen during training**, representing **genuine zero-shot generalization**.
>     * To the best of our knowledge, current commercial APIs and other methods rely on **condition-specific or task-specific finetuning**. They **do not support zero-shot semantic generalization** to unseen effects within a *single unified model*.
>     * In contrast, **Video-As-Prompt** is the **first unified framework** to achieve this by treating the reference video as a prompt, rather than relying on separate finetuned models per condition.

---

> ### Author Response · Authors · 2025-11-19
> **Response to Reviewer v4tD (3/5)**
>
> > ### Weakness 4, Question 2: Explanation About the Ablation Quantitative Results
>
> We appreciate the opportunity to clarify our ablation results.
>
> 1.  **Metric Sensitivity in Video Evaluation.** We agree that the improvements in Table 2 may appear marginal if viewed solely through absolute numbers. This is primarily because **current quantitative metrics are not sufficiently sensitive to perceptual differences in video generation quality**. For instance, on the widely used VBench leaderboard (https://huggingface.co/spaces/Vchitect/VBench_Leaderboard), the community generally agrees that **Veo 3 > Wan 2.1 > CogVideoX-5B** in quality. However, standard metrics fail to capture this distinct ranking:
>
>     | Model | Motion Smoothness ↑ | Dynamic Degree ↑ | Aesthetic Quality ↑ | Overall Consistency (Clip Score) ↑ |
>     | :--- | :---: | :---: | :---: | :---: |
>     | **Veo 3** | 99.16% | 72.43% | 63.81% | 27.88% |
>     | **Wan 2.1** | 96.92% | 94.35% | 61.53% | 27.44% |
>     | **CogVideoX-5B** | 97.20% | 69.51% | 61.88% | 27.65% |
>
>     Despite the clear perceptual gap, the **absolute metric differences are negligible**. This illustrates a broader industry challenge: standard metrics often struggle to distinguish between high-quality models.
>
> 2.  **Our Semantic-Alignment Metric.** To address this, we introduced a **VLM-based semantic-alignment score** (Sec. 4.2) to directly measure the consistency between reference and generated videos. In Table 2, this metric ranges from **55.99 to 70.44**, providing a **much clearer separation** between variants that aligns with their perceptual quality. To validate this, **Table 5** compares our model against baselines using:
>     * Semantic-alignment score (Gemini-1.5-Pro).
>     * Semantic-alignment score (GPT-4o).
>     * User Preference Rate.
>
>     We observe **high trend consistency** among these three signals, confirming the **reliability** of our semantic-alignment metric even when traditional metrics show smaller deltas.
>
> 3.  **Interpretation.** We interpret the ablation results based on their **consistent relative improvements** and the strong correlation with human preference, rather than raw magnitudes of insensitive metrics.

---

> ### Author Response · Authors · 2025-11-19
> **Response to Reviewer v4tD (4/5)**
>
> > ### Weakness 5, Question 3: More Discussion and Potential Solution Regarding the Performance Drop under Mismatched Captions/Objects
>
> 1.  **Frequency of Issue.** We appreciate the concern. While noticeable performance drops occur in challenging cases, this is not representative of typical usage. As shown in our supplementary material ([https://video-as-prompt.github.io/](https://video-as-prompt.github.io/)), the model often successfully transfers semantics even when reference and target videos differ substantially in structure. We highlighted these limitations in **Figure 14** to be transparent about failure modes, which typically occur when (i) captions are low-quality/inconsistent, or (ii) object structures differ drastically.
>
> 2.  **Potential Cause: Ambiguous Semantic Anchors.** We have expanded the analysis around **Lines 1129–1133**. Our model typically:
>     * First, relies on the **most critical shared semantics** in the captions to establish a **coarse "semantic anchor."**
>     * Then, uses the **reference video** to fill in fine-grained details (appearance, motion, style).
>
>     Our current reliance on **descriptive captions** (which lack referential grounding) can cause this anchor to become ambiguous when the description is overly complex or under-specified, confusing the model's semantic selection.
>
> 3.  **Potential Solution: Instruction-Style Captions.** As stated in **Sec. 5 (Lines 481–485)**, we hypothesize that **instruction-style captions** (e.g., *“please follow the Ghibli style in the reference video”*) would better capture the **intended control signal** and improve robustness, even when objects differ explicitly.
>
> 4.  **Zero-Shot Evidence.** To support this, we performed a **zero-shot test** using our current checkpoint (trained *only* on descriptive captions). We compared:
>     > * **Original Descriptive Caption:** "A chestnut-colored horse stands on a grassy hill against a backdrop of distant, snow-dusted mountains. The horse begins to inflate, its defined, muscular body swelling and rounding into a smooth, balloon-like form while retaining its rich, brown hide color. Without changing its orientation, the now-buoyant horse lifts silently from the ground. ... The camera remains completely static..."
>
>     > * **New "Weak" Instruction-Style Caption:** "The horse begins to inflate, become a balloon-like form, and lift silently from the ground."
>
>     Despite never seeing such "Weak" Instruction-Style captions during training—and despite these captions only mentioning the core semantics without describing object appearance or camera constraints, the model **stably transferred the intended semantics** [anonymous link: [https://anonymous.4open.science/r/Video-As-Prompt-Review-CA61/rebuttal/weak\_instruction\_example.mp4](https://anonymous.4open.science/r/Video-As-Prompt-Review-CA61/rebuttal/weak_instruction_example.mp4)]. This provides **strong initial evidence** that instruction-style captions are compatible with our framework and represent a promising direction for future work.

---

> ### Author Response · Authors · 2025-11-19
> **Response to Reviewer v4tD (5/5)**
>
> > ### Question 4: Open-source VAP-Data
>
> **Yes.** As stated in **Sec. 7 (Reproducibility Statement)** and our anonymous [code repository](https://anonymous.4open.science/r/Video-As-Prompt-Review-CA61), **we commit to releasing the full VAP-Data dataset, along with all training code, inference code, and model checkpoints**.
>
> We believe that open access to **large-scale, semantically controlled video data** is crucial for the community. By releasing these resources, we aim to facilitate reproducibility, enable fair comparisons, and contribute a solid foundation for future research in unified semantic control.

---

### Official Review · Reviewer_sfsG · 2025-10-28

**Soundness:** 2
**Presentation:** 2
**Contribution:** 2
**Rating:** 2
**Confidence:** 3

**Summary:**

The paper presents a unified framework for efficiently compressing large-scale video datasets into compact synthetic ones while preserving both spatial and temporal dynamics. Unlike prior two-stage video distillation methods such as VDSD and IDTD, which are computationally heavy and fail to model motion coherence, the proposed approach introduces a uni-level optimization framework enhanced by a Temporal Saliency-Guided Filter (TSGF). This filter leverages inter-frame differences to guide the distillation process, adaptively constraining optimization to retain key motion cues and suppress redundant frames, and further employs temporally guided augmentation to enhance diversity without breaking temporal continuity. Extensive experiments on benchmarks including MiniUCF, HMDB51, Kinetics-400, and SSv2 demonstrate state-of-the-art performance and efficiency gains, achieving significant improvements under extreme compression while maintaining strong temporal consistency, thus offering a scalable and effective solution for video dataset distillation.

**Strengths:**

1.	The paper introduces a uni-level video dataset distillation framework that eliminates the complexity of prior bi-level methods such as VDSD and IDTD. This design effectively reduces computational time and memory consumption while maintaining high performance.
2.	A novel TSGF module is proposed to guide the distillation process by computing inter-frame differences, which helps preserve and enhance temporal coherence. This ensures that essential motion information is retained throughout the optimization process.
3.	The authors further design a temporal-aware data augmentation technique that enhances the diversity of synthetic videos without disrupting their temporal consistency.
4.	The framework demonstrates robustness across datasets of different scales and domains, offering a scalable and effective paradigm for video dataset distillation.

**Weaknesses:**

1.	Some figures (e.g., Fig. 1, Tab. 2) contain typographical errors and could benefit from clearer annotations or more consistent formatting to enhance readability.
2.	The Temporal Saliency-Guided Filter (TSGF) relies on simple inter-frame differences to estimate temporal importance, which may be sensitive to noise, static scenes, or camera motion. A discussion of robustness or potential failure cases would strengthen the work.
3.	The reference format are inconsistencies.
4.	Although the paper claims lower computational cost, it does not provide detailed runtime or memory comparisons under consistent hardware conditions. Quantitative evidence for scalability would make the efficiency claim more convincing.
5.	The claimed efficiency of the uni-level framework may be overstated, as bi-level methods can reuse pre-trained image models or distilled datasets, potentially achieving lower cost and stronger spatial representations.

**Questions:**

See Weakness.

---

> ### Author Response · Authors · 2025-11-14
> **Mis-uploaded Review**
>
> Dear Reviewer sfsG,
>
> Thank you very much for taking the time to review our submission.
>
> We noticed that the review attributed to you under our paper seems to discuss **temporal saliency–guided dataset distillation**, which is not part of our manuscript. It appears to match another ICLR submission.
>
> * Review shown under our paper:
>   [https://openreview.net/forum?id=8FihPljvWf&noteId=4BWbnYsXpt](https://openreview.net/forum?id=8FihPljvWf&noteId=4BWbnYsXpt)
> * Review shown under the other submission (likely about our paper):
>   *Temporal Saliency-Guided Distillation: A Scalable Framework for Distilling Video Datasets* —
>   [https://openreview.net/forum?id=vYqAuAuV1v&noteId=OLijiGXsUc](https://openreview.net/forum?id=vYqAuAuV1v&noteId=OLijiGXsUc)
>
> This might simply be an upload mix-up. If so, would you mind taking a quick look and updating the review? In the meantime, we will prepare our rebuttal based on the review that appears intended for our paper.
>
> We truly appreciate your effort and help.
>
> Sincerely,
>
> **Video-As-Prompt Authors**

---

> ### Author Response · Authors · 2025-11-19
> **Response to Reviewer sfsG (1/3)**
>
> We sincerely thank the reviewer for the insightful comments and recognition of this work, especially for acknowledging the **soundness and novelty** of our video-as-prompt reformulation, in-context MoT generation, and VAP-Data construction.
>
> **Note on the Review:** We noticed that the review uploaded under our paper appears to be for another submission (likely "Temporal Saliency-Guided Distillation..."). Since we have not received a correction, we have proceeded to **prepare our rebuttal based on the review that appears intended for our submission** (the content covering MoT computation, added Omni-Effects comparison, and added zero-shot quantitative experiments).
>
>
>
> > ### Weakness 1: The Mixture-of-Transformers (MoT) architecture... nearly doubles the total parameter count and substantially increases computational cost.
>
> We appreciate the concern regarding scalability and clarify that the MoT structure primarily increases the parameter count. The **additional computational cost mainly stems from the in-context generation task itself, not from the dual-transformer (MoT) structure**.
>
> 1.  **Clarification: Compute vs. Memory.** The primary overhead of our MoT architecture is **memory, not compute**.
>     * **Memory:** The increased **memory usage** stems from introducing an additional in-context DiT expert.
>     * **Compute:** The increased **compute** (and thus inference time) is a fundamental property of the **in-context generation task**, which requires processing a longer, concatenated sequence (reference_caption, reference_video, target_caption, target_video). This cost is **not unique to MoT**; all in-context baselines (e.g., $u_{\Theta}^{s}$, $u_{\Theta}^{sl}$ in Table 2) incur this overhead.
>     * **MoT vs. Single-DiT Compute:** For the same in-context task, the dominant costs (full attention and FFNs) are **theoretically the same FLOPs** in our MoT as in a single-DiT. The MoT architecture primarily changes **parameterization and memory footprint**, but does not inherently increase the computational cost of the in-context setup.
>
> 2.  **Parameter-Efficient Variants and Scope.** We acknowledge the need for efficiency and have explored **parameter- and memory-friendly variants** in **Appendix F, Table 7** (*In-Context Expert Transformer Layer Distribution*).
>     * While our main version uses a full expert copy for strongest performance (used in our CogVideoX version), we show that **distributed / interval replication** (used in our Wan2.1 version) achieves competitive control while **significantly reducing memory consumption** and lowering effective compute.
>     * A systematic exploration of highly-optimized MoT designs (e.g., pruning, sparse attention) is an important, but **orthogonal, direction** to our core contribution: establishing a *unified framework for semantic control*. We explicitly leave these optimizations to future work (Appendix E.3).

---

> ### Author Response · Authors · 2025-11-19
> **Response to Reviewer sfsG (2/3)**
>
> > ### Weakness 2: More Comparison with Omni-Effects
>
> 1.  **New Quantitative Comparison.** We fully agree that adding a detailed comparison with Omni-Effects strengthens our claims on unification and generalization. We added quantitative results using the **multi-VFX variant of Omni-Effects**, evaluated under the **same protocol as Sec. 4.3**.
>
> 2.  **Empirical Findings.** Under our evaluation, Omni-Effects achieves multi-semantic control, but its reliance on **semantic-specific overfitting in parameter space** leads to weaker **generalization** and degrades **general video generation capability**. This is reflected by its **consistently lower average metrics** compared to our method:
>
> | Model | Clip Score $\uparrow$ | Motion Smoothness $\uparrow$ | Dynamic Degree $\uparrow$ | Aesthetic Quality $\uparrow$ | Semantic Alignment Score $\uparrow$ |
> | :--- | :---: | :---: | :---: | :---: | :---: |
> | Omni-Effects (multi-VFX) | 23.56 | 98.31 | 72.92 | 55.14 | 67.38 |
> | **Video-As-Prompt (VAP)** | **24.13** | **98.59** | **77.08** | **57.71** | **70.44** |
>
> 3.  **Differences in Unification and Scalability.** Omni-Effects achieves multi-condition control via **multiple, separate LoRA modules** for different semantics. This design has two major drawbacks compared to VAP:
>     * It cannot support **zero-shot semantic control** since each new semantic requires a trained LoRA.
>     * It is **not scalable to arbitrary numbers of conditions** with a fixed parameter budget, as each new semantic requires additional LoRA parameters.
>     * In contrast, VAP uses **a single set of parameters** (the MoT expert) trained on arbitrarily many semantics, enabling true **zero-shot generalization**.
>
> 4.  **Reason for Omission.** We strongly agree this comparison improves the paper. We originally omitted a quantitative comparison, providing only a **qualitative discussion (Sec. 2, lines 187–190)**, because Omni-Effects is a **concurrent, not-yet-peer-reviewed** work. We have now **added the requested quantitative comparison** to make the paper more complete.

---

> ### Author Response · Authors · 2025-11-19
> **Response to Reviewer sfsG (3/3)**
>
> > ### Weakness 3: Added Quantitative Results of the Zero-shot Experiments
>
> 1.  **New Quantitative Evaluation.** We agree that quantitative validation of the zero-shot setting is critical. We evaluated the default **Video-As-Prompt (CogVideoX)** configuration on the four **unseen semantic conditions** shown in Figure 7, using the *same evaluation protocol* as Sec. 4.3. For a fair comparison, we generate **12 test samples per unseen semantic**, so that the **total number of evaluation samples (48)** matches that of the main experiment.
>
> 2.  **Consistency on Seen vs. Unseen Semantics.** Quantitatively, the metrics on these four **unseen** semantics are comparable to those on the **seen** semantics reported in Table 1, demonstrating that our model maintains **consistent performance in both settings**. This is enabled by our **unified semantic representation** (reference video as prompt) and **unified framework design** (in-context MoT), allowing the model to generalize beyond the training semantics in a zero-shot manner.
>
> | Model | Clip Score $\uparrow$ | Motion Smoothness $\uparrow$ | Dynamic Degree $\uparrow$ | Aesthetic Quality $\uparrow$ | Semantic Alignment Score $\uparrow$ |
> | :--- | :---: | :---: | :---: | :---: | :---: |
> | **VAP (seen semantics)** | 24.13 | 98.59 | 77.08 | 57.71 | 70.44 |
> | **VAP (zero-shot semantics)** | 24.07 | 98.48 | 79.17 | 57.65 | 70.08 |

---

### Official Review · Reviewer_KiRJ · 2025-11-01

**Soundness:** 3
**Presentation:** 2
**Contribution:** 3
**Rating:** 6
**Confidence:** 3

**Summary:**

This paper presents Video-As-Prompt (VAP), a unified framework that treats a reference video as a prompt for semantically controlled video generation. It enables multiple controls such as concept, style, motion, and camera within a single model, achieving strong generalization and competitive performance.

**Strengths:**

1. The paper proposes a unified “video-as-prompt” paradigm that reframes semantic control in video generation as an in-context learning problem, offering a clear conceptual advance over task-specific approaches.
2. The temporally biased RoPE effectively avoids pixel-level copying between reference and target videos, leading to more robust semantic alignment.
3. The Mixture-of-Transformers design enables plug-and-play integration with existing video diffusion transformers while preventing catastrophic forgetting.

**Weaknesses:**

Major:
1. The paper lacks a theoretical analysis explaining why in-context learning via Mixture-of-Transformers effectively transfers semantic patterns.
2. The proposed temporally biased RoPE is only justified empirically, without an ablation or analytical study on the optimal bias magnitude.
3. The inference cost roughly doubles due to the dual-transformer structure, yet efficiency and scalability trade-offs are not thoroughly studied.

Minor:
1. The semantic diversity in VAP-Data is constrained to four categories, leaving out high-level semantics such as narrative or causal events; but the constructed dataset is the largest to date for semantically controlled video generation and provides a valuable foundation for future research.
2.  The work does not analyze failure cases quantitatively, leaving unclear when and why semantic leakage or identity loss occurs.

Some Typo issues:
1. Line 131: Add a space in "architectures(Singer et al., 2022; …)" to "architectures (Singer et al., 2022; …)".
2. Line 157: Add a space in "Concept(Liu et al., 2025; …)" to "Concept (Liu et al., 2025; …)".
3. Line 160: Add spaces in "Camera Movement(He et al., 2024; …)" and "Motion(Zhao et al., 2024; …)" to "Camera Movement (He et al., 2024; …)" and "Motion (Zhao et al., 2024; …)".
4. Lines 239 and 390: Ensure consistent usage of “fine-tuning” instead of “finetuning”.
5. Line 300: Add a space in "480×720(832)" to "480×720 (832)".
6. Line 321: Add a space in "resource(see Sec. 3.2 …)" to "resource (see Sec. 3.2 …)".
7. Line 1026: Correct the typo "metics" to "metrics".
8. Line 1240: Correct capitalization "LoRa" to "LoRA".

**Questions:**

Refer to Major part in Weaknesses.

---

> ### Author Response · Authors · 2025-11-19
> **Response to Reviewer KiRJ (1/6)**
>
> We express our sincere gratitude to the reviewer for the insightful comments and the recognition of our work. We especially appreciate the acknowledgment of our approach's novelty, effectiveness, and superior performance. We have addressed the specific concerns raised by the reviewer as detailed below.
>
> > ### Major Weakness 1: Lack of a theoretical analysis explaining why in-context learning via Mixture-of-Transformers effectively transfers semantic patterns.
>
> We thank the reviewer for suggesting this theoretical perspective. Below, we provide a formal intuition for why our framework effectively disentangles and transfers semantics.
>
> **1. Problem Formulation**
> Assume each training pair is generated from a **shared latent semantic** $s$ (e.g., "a horse becoming a Minecraft character") and different appearance layouts $l_r, l_t$ for the reference and target:
>
> $$
> (R_{video}, T_{video}) \sim p(\cdot \mid s, l_r, l_t), \quad
> (R_{text}, T_{text}) \sim p(\cdot \mid s).
> $$
>
> In Video-As-Prompt, we concatenate all tokens into a single sequence:
> $$
> X = (R_{text}, R_{video}, T_{text}, T_{video}).
> $$
> We then apply temporally biased RoPE and run a stack of MoT blocks with **full attention over all tokens**. At a given layer, for a target token $x_i^T$, the self-attention mechanism computes:
>
> $$\alpha_{ij} = softmax_j(q_i^\top k_j), \quad
> x_i^{new} = \sum_j \alpha_{ij} v_j,
> $$
>
> where the index $j$ ranges over both reference and target tokens.
>
> **2. Content-Based Semantic Matching**
> Because our temporally biased RoPE assigns **disjoint temporal indices** to the reference and target videos, there is **no positional prior** forcing frame-aligned copying. In addition, the reference video and the target video each have their own appearance and layout. Consequently, the attention scores $q_i^\top k_j$ are dominated by **semantic content similarity** in the learned feature space.
>
> Under the data assumption above, tokens in the reference region that encode the same semantic $s$ as the current target token appear with **systematically higher covariance** across the dataset. Minimizing the diffusion loss pushes the attention mechanism to approximate a **kernel regression** over these statistics:
>
> $$
> \alpha_{ij} \propto \exp(q_i^\top k_j) \approx \mathcal{K}(\phi(x_i^T), \phi(x_j^R)),
> $$
> where $x_j^R$ are reference tokens and $\phi$ is the learned feature map. In expectation, the optimal attention pattern assigns **significantly larger weights** to reference positions that encode the same semantic pattern $s$.
>
> **3. Semantic Extraction and Injection**
> The update for a target token can be decomposed into a reference component and a target component:
>
> $$
> x_i^{\text{new}} = \sum_{j \in R} \alpha_{ij} v_j^R + \sum_{j \in T} \alpha_{ij} v_j^T
> $$
>
> We can define the first term as a **semantic summary** of the reference for this position, $\Psi_s(R)$, which aggregates how the semantic $s$ is realized in the reference (e.g., style, motion, camera pattern). Stacking layers yields a mapping of the form:
>
> $$
> \Phi_T^{(L)} \approx G(\Phi_T^{(0)}, \Psi_s(R)),
> $$
>
> where $\Phi_T^{(L)}$ are the final target features. Thus, at each layer, the attention mechanism **extracts a semantic descriptor from the reference and injects it into the target features**, ensuring the denoising trajectory is driven effectively by $\Psi_s(R)$.
>
> **4. Role of MoT with a Frozen Backbone**
> The MoT expert, initialized from the pretrained DiT, is trained such that:
> * Reference tokens are encoded into the semantic control signal $\Psi_s(R)$.
> * Attention between the expert and the frozen backbone **modulates** the target features along directions specified by $\Psi_s(R)$.
>
> Since gradients arise from the **joint denoising of $(R_{video}, T_{video})$ conditioned on the same $s$**, the expert learns to **reuse semantic directions present in the reference representation** to steer the frozen generator, rather than learning independent rules for each caption.

---

> ### Author Response · Authors · 2025-11-19
> **Response to Reviewer KiRJ (2/6)**
>
> > ### Major Weakness 2: The proposed temporally biased RoPE is only justified empirically, without an ablation or analytical study on the optimal bias magnitude.
>
> We appreciate this constructive suggestion. We have added a detailed explanation of our design principles and a new ablation study to demonstrate the robustness of our approach to this hyperparameter.
>
> 1.  **Design Principle.** Our choice of the temporal offset ($\Delta$) is governed by the core principle of **avoiding temporal overlap between reference and target videos while keeping their spatial positions aligned**. In our default CogVideoX and Wan2.1 versions, RoPE is applied only to video tokens. Let the temporal length of both the reference and target videos be $T$. We assign the target video temporal indices $[0, T-1]$ and set $\Delta = T$, so the reference video uses the indices $[-T, -1]$. This temporally biased RoPE:
>     * Prevents any artificial one-to-one pixel mapping prior between reference and target frames.
>     * Intuitively aligns with the concept of **in-context video generation**, where the reference serves as a context rather than as frame-aligned supervision.
>
>     As shown in **Table 2 (Position Embedding Design)** and **Fig. 5(c,d)**, this design empirically yields superior performance by avoiding spurious mapping priors.
>
> 2.  **Hyperparameter Choice and Ablations.** We further clarify the choice of $\Delta$ and have added new ablations on its sensitivity. We tested four variants against our default CogVideoX variant:
>     * **A1 (larger, multiple of $T$):** $\Delta = 2T$
>     * **A2 (larger, non-multiple):** $\Delta = 1.5T$
>     * **A3 (smaller, partial overlap):** $\Delta = 0.5T$
>     * **A4 (opposite sign):** $\Delta = -T$ (placing the reference *after* the target)
>
>     As shown in the table below, we found that the model is **generally insensitive** to the exact value of $\Delta$, as long as the temporal position ranges of the reference and target videos do not overlap. The quantitative metrics for A1, A2, and A4 remain nearly identical to our default.
>
> 3.  **Robustness and Takeaway.** The only case with a clear performance degradation is **A3 ($\Delta = 0.5T$)**. Here, the temporal ranges partially overlap, which reintroduces the erroneous priors that implicitly encourage a non-existent pixel-wise correspondence, thus hurting performance. This confirms that the key requirement is simply that the **reference and target temporal positions remain disjoint**.
>
> | Variant | Text CLIP Score $\uparrow$ | Motion Smoothness $\uparrow$ | Dynamic Degree $\uparrow$ | Aesthetic Quality $\uparrow$ | Semantic Alignment Score $\uparrow$ |
> | :--- | :---: | :---: | :---: | :---: | :---: |
> | **A1 ($\Delta = 2T$)** | 24.05 | 98.63 | 75.00 | 57.68 | 70.23 |
> | **A2 ($\Delta = 1.5T$)** | 24.07 | 98.57 | 79.17 | 57.62 | 70.41 |
> | **A3 ($\Delta = 0.5T$, overlap)** | 23.62 | 98.52 | 72.92 | 57.25 | 69.24 |
> | **A4 ($\Delta = -T$)** | 24.03 | 98.55 | 77.08 | 57.70 | 70.15 |
> | **VAP ($\Delta = T$, default)** | **24.13** | **98.59** | **77.08** | **57.71** | **70.44** |

---

> ### Author Response · Authors · 2025-11-19
> **Response to Reviewer KiRJ (3/6)**
>
> > ### Major Weakness 3: The inference cost roughly doubles due to the dual-transformer structure, yet efficiency and scalability trade-offs are not thoroughly studied.
>
> We thank the reviewer for this comment. We would like to clarify the source of the computational cost and highlight our efficiency studies.
>
> 1.  **Inference Cost = In-Context Compute + MoT Memory.**
>     The observed 2× inference cost primarily stems from the **in-context generation task itself**, not the specific dual-transformer (MoT) architecture.
>     * **Compute:** Moving from a standard DiT to VAP involves changing the input from a single sequence (target_caption, target_video) to a concatenated **in-context sequence** (reference_caption, reference_video, target_caption, target_video). Any method processing this concatenated sequence—including our single-branch finetuning baselines ($u_{\Theta}^{s}$ and $u_{\Theta}^{sl}$ in Tab. 2)—would incur this cost. In our MoT, the dominant cost (full attention + FFNs) is **theoretically identical in FLOPs** to a single-DiT setup, as we simply split the FFN parameters between two experts. Thus, MoT changes the *parameterization*, not the inherent *compute*.
>     * **Memory:** The primary overhead of the MoT design is **memory**, as we maintain an additional in-context expert branch.
>
> 2.  **Quantitative Ablations on Efficiency.**
>     We have quantitatively studied these trade-offs in **Appendix F, Table 7**:
>     * **Parameter-Efficient Distributions:** We show that while a full expert copy (finally adopted by our CogVideoX version) offers maximum control, **interval/partial replication** of layers (finally adopted by our Wan2.1 version) significantly reduces memory usage and the number of full attention layers, delivering competitive performance with lower computational cost.
>     * **Future Work:** As noted in Appendix E.3, further optimizations (e.g., sparse attention, aggressive pruning) are orthogonal to our contribution of a unified control framework and are left for future work.

---

> ### Author Response · Authors · 2025-11-19
> **Response to Reviewer KiRJ (4/6)**
>
> > ### Minor Weakness 1: Semantic diversity in VAP-Data is constrained to four categories.
>
> We acknowledge the limited diversity of the dataset but would like to emphasize its unprecedented scale relative to existing resources.
>
> While VAP-Data is limited to four categories with 100 semantic condition types, it is the **largest available dataset** for semantic control in video generation. The previous largest dataset, **Open-VFX** ([https://huggingface.co/datasets/sophiaa/Open-VFX](https://huggingface.co/datasets/sophiaa/Open-VFX)), contains only 559 samples across 14 conditions, which mainly only focuses on concept category. In contrast, VAP-Data expands the scale by over **178×**, offering **100k+ videos across 100 semantic conditions**. This provides **the first practical, training-friendly data foundation** for *unified* semantic-controlled video generation. We are committed to releasing this dataset to promote further research in the community.

---

> ### Author Response · Authors · 2025-11-19
> **Response to Reviewer KiRJ (5/6)**
>
> > ### Minor Weakness 2: Lack of quantitative analysis of failure cases.
>
> 1.  **Observed Failure Patterns.** We agree that analyzing failures is crucial. Empirically, we observe that **semantic leakage and identity loss** most often occur when:
>     * **Captions are inaccurate:** The prompt does not correctly describe the reference semantics.
>     * **Reference quality is low:** The reference video has unclear semantics, strong jitter, or is overly static.
>
>     We have provided concrete qualitative examples of these modes in **Figure 14**.
>
> 2.  **Limitation.** A key challenge is that **quantitatively measuring the quality of conditioning signals** (i.e., caption accuracy and reference semantic clarity) remains an open problem. Designing robust metrics to explain *when and why* failures occur requires jointly modeling caption accuracy and visual quality, which is beyond the current scope. We will closely follow progress in this area and plan to extend our framework with quantitative failure diagnostics in future work.

---

> ### Author Response · Authors · 2025-11-19
> **Response to Reviewer KiRJ (6/6)**
>
> > ### Typo Issue
>
> We are very grateful for the reviewer's detailed inspection, which has been of immense help in improving the quality of our paper. We have corrected all the identified typos in the revised manuscript.

---

### Official Review · Reviewer_1zha · 2025-11-03

**Soundness:** 3
**Presentation:** 3
**Contribution:** 3
**Rating:** 6
**Confidence:** 3

**Summary:**

This paper proposes Video-As-Prompt (VAP), a unified framework for semantic-controlled video generation. Instead of pixel-aligned control (e.g., depth, pose), the method uses reference videos as prompts to transfer concepts, styles, motions, and camera behaviors. A Mixture-of-Transformers (MoT) design augments a frozen video DiT with a parallel expert transformer, enabling plug-and-play in-context conditioning while avoiding catastrophic forgetting. A temporally-biased RoPE removes false spatial correspondences between reference and target tokens. The authors also introduce VAP-Data, a 100K-pair synthetic benchmark across 100 semantic categories. Experiments show strong semantic alignment, competitive with commercial models, and notable zero-shot generalization to unseen effects.

**Strengths:**

+ The paper introduces a clean and unified perspective by treating a reference video as a semantic prompt, avoiding the fragmented control paradigms (e.g., pose/depth-specific pipelines) used in prior work.

+ The MoT structure and temporally-biased RoPE are well-motivated and demonstrated to be effective in preventing forgetting and cross-token interference; ablations support the design choices.

+ Strong qualitative performance across multiple semantic axes (concept, style, motion, camera intent), with notable zero-shot generalization and performance competitive with proprietary systems.

**Weaknesses:**

- The training data is largely synthetic and template-driven, which may limit generalization to real-world video distributions; robustness to natural, diverse videos is not extensively evaluated.

- The MoT architecture increases compute cost and memory footprint, making the method relatively heavy compared to lightweight or plug-in control modules.

- The method assumes reasonably descriptive captions for reference and target videos; behavior under noisy or under-specified captions remains insufficiently analyzed.

**Questions:**

- How does the method perform on noisy, compressed, or hand-held reference videos? Any robustness evaluation or failure cases that could be shared?

- The paper mentions instruction-style prompting as potentially beneficial. Have the authors quantified improvements when using such synthetic instructions versus descriptive captions?

- Is the MoT design amenable to parameter-efficient variants (e.g., partial expert layers or shared attention blocks)? If so, did the authors explore such configurations?

---

> ### Author Response · Authors · 2025-11-19
> **Response to Reviewer 1zha (1/6)**
>
> We sincerely thank the reviewer for the insightful comments and recognition of this work, especially for acknowledging the soundness and novelty of our unified semantic video generation framework based on "video-as-prompt" reformulation. We have clarified the below points and added the needed quantitative and qualitative experiments.
>
>
>
> > ### Weakness 1: Synthetic Data Limitations
>
>
> We thank the reviewer for this important point. We agree with the synthetic data limitation. As discussed in Sec. 5 Conclusion (Limitation and Future Works) and Appendix D Dataset, we chose synthetic data construction because creating **a large-scale, real-world, semantically controlled video dataset** is *infeasible* at this exploratory stage due to **extremely high annotation costs**.
>
> However, we believe our VAP-Data and the Video-As-Prompt framework offer significant contributions to the community in three key areas:
>
>
> 1. **Data Contribution.** Although VAP-Data is synthetic, it is, to the best of our knowledge, the **largest available dataset** for semantic control in video generation. The previous largest dataset, **Open-VFX** ([https://huggingface.co/datasets/sophiaa/Open-VFX](https://huggingface.co/datasets/sophiaa/Open-VFX)), is also synthetic and contains only 559 samples across 14 semantic conditions. In contrast, VAP-Data expands the scale by over **178×, offering more than 100k videos across 100 semantic conditions**. This provides **the first practical, training-friendly data foundation** for *unified* semantic-controlled video generation.
>
> 2.  **Research Contribution.** Our primary goal is to study **how to transfer abstract semantics from a reference video to a target video** under a unified formulation. Our *Video-As-Prompt* approach, combined with the MoT expert and temporally-biased RoPE, validates this core idea. To our knowledge, this is the **first work in the community to validate the "video-as-prompt" concept for unified semantic control**, filling an important gap in controllable video generation.
>
> 3.  **Generalization to Real-World Data.** To further assess generalization, we **added zero-shot experiments** to show that although VAP is trained solely on synthetic data, it **generalizes well to real-world reference videos and complex composite semantics**. For example, our model can reproduce:
>
>       * The complex semantics of a character raising both arms with their back to an explosion (e.g., the "Iron Man" movie) [anonymous link: [https://anonymous.4open.science/r/Video-As-Prompt-Review-CA61/rebuttal/real\_video\_example\_1.mp4](https://anonymous.4open.science/r/Video-As-Prompt-Review-CA61/rebuttal/real_video_example_1.mp4)].
>       * The famous real-world "Michael Bay shot" (a rapid, low-angle, orbiting shot) [anonymous link: [https://anonymous.4open.science/r/Video-As-Prompt-Review-CA61/rebuttal/real\_video\_example\_2.mp4](https://anonymous.4open.science/r/Video-As-Prompt-Review-CA61/rebuttal/real_video_example_2.mp4)].
>       * Complex composite effects, such as transforming into a robot (i.e., "Transformer") while the environment simultaneously turns to winter [anonymous link: [https://anonymous.4open.science/r/Video-As-Prompt-Review-CA61/rebuttal/multi\_vfx\_example.mp4](https://anonymous.4open.science/r/Video-As-Prompt-Review-CA61/rebuttal/real_video_example_3.mp4)].
>
> Finally, we sincerely agree with the reviewer on the importance of a large-scale, real-world, semantic-controlled video dataset. As stated in our limitations (Line 480), we leave this as an important direction for future work.

---

> ### Author Response · Authors · 2025-11-19
> **Response to Reviewer 1zha (2/6)**
>
> > ### Weakness 2: The MoT architecture increases compute cost and memory footprint, making the method relatively heavy compared to lightweight or plug-in control modules.
>
>
> We thank the reviewer for this important point. We would like to clarify the distinction between the computational and memory overheads of our approach.
>
> 1.  **Clarification: Compute vs. Memory.** The primary overhead of our MoT architecture is **memory, not compute**.
>     * **Memory:** The increased **memory usage** stems from introducing an additional in-context DiT expert.
>     * **Compute:** The increased **compute** (and thus inference time) is a fundamental property of the **in-context generation task itself**, which requires processing a longer, concatenated sequence (reference_caption, reference_video, target_caption, target_video). This computational cost is **not unique to MoT**. As shown in Table 2, our single-branch baselines ($u_{\Theta}^{s}$, $u_{\Theta}^{sl}$ ) that also perform in-context generation incur this same overhead.
>     * **MoT vs. Single-DiT Compute:** For the same in-context task, the dominant computational costs (joint full attention and FFNs) are **theoretically the same FLOPs** in our MoT as they would be in a single-DiT model. The MoT architecture primarily changes the **parameterization and memory footprint** of the in-context setup, but does not inherently add to its computational cost.
>
> 2.  **Parameter-Efficient Variants and Scope.** We acknowledge the importance of efficiency and have already explored **more parameter- and memory-friendly variants** in **Appendix F (Ablation Study)**, specifically **Table 7** (*In-Context Expert Transformer Layer Distribution*).
>     * While our main CogVideoX version uses a full expert copy for maximum performance, we show that **distributed / interval replication** (used in our Wan2.1 version) achieves competitive control while **significantly reducing memory consumption** and lowering effective compute.
>     * We consider a systematic exploration of highly-optimized, lightweight MoT designs (e.g., aggressive sharing, pruning, or sparse attention) to be an important, but **orthogonal, direction** to our paper's core contribution: establishing a *unified framework for semantic control*. As discussed in Appendix E.3 (Efficiency), we explicitly leave these efficiency-focused optimizations as a promising avenue for future work.

---

> ### Author Response · Authors · 2025-11-19
> **Response to Reviewer 1zha (3/6)**
>
> > ### Weakness 3: Insufficient Behavior Analysis Under Noisy or Under-Specified Captions
>
> Thank you for pointing out the importance of analyzing noisy and under-specified captions.
>
>
> We thank the reviewer for highlighting the importance of analyzing model behavior with noisy and under-specified captions.
>
> 1.  **Existing Analysis.** We agree this is a key point. As shown in **Fig. 14** (Appendix) and discussed in **Lines 1129-1133**, we have already conducted qualitative tests with **low-quality, semantically inaccurate captions**. We observe that inaccurate descriptions in the reference caption clearly degrade semantic consistency. For example, in Fig. 14, the "bad" reference caption incorrectly describes the effect as “water pours over” instead of “liquid metal emerging on the body,” which misleads the model and causes it to fail to generate the characteristic properties of liquid metal.
>
> 2.  **Further Analysis and Clarification.** To further elaborate, we have clarified the mechanism behind this failure. We empirically observe that Video-As-Prompt typically:
>     * First, relies on the **most critical shared semantics** in the reference and target captions to establish a **coarse-level "semantic anchor"** for the concept to be transferred.
>     * Then, it leverages the **reference video** to fill in the finer-grained details (e.g., stylistic attributes, motion patterns, or camera speed).
>
>     When captions are noisy, under-specified, or semantically misaligned, this "anchor" becomes ambiguous. This ambiguity directly **impairs the model's semantic selection** and leads to the observed performance degradation. We have incorporated this more detailed analysis on caption robustness into the revised manuscript (lines 1185-1187, lines 1213-1219).

---

> ### Author Response · Authors · 2025-11-19
> **Response to Reviewer 1zha (4/6)**
>
> > ### Question 1: How does the method perform on noisy, compressed, or hand-held reference videos?
>
>
> We thank the reviewer for this important practical question. In response, we have **added a new ablation study where we explicitly degrade the quality of the reference videos** to simulate real-world noise.
>
> Specifically, for each reference video in our test set, we apply two separate degradations:
> 1.  **Noise Injection:** We inject **Gaussian noise** ($\sigma = 0.05$) to simulate sensor noise or mild compression artifacts.
> 2.  **Low Resolution:** We **spatially downsample** the reference videos to mimic low-bitrate or heavily compressed inputs.
>
> We then evaluate these degraded-reference inputs using the same five quantitative metrics from our main experiments.
>
> **Observed Impact on Performance:** As summarized in the table below, **degrading the reference video quality consistently lowers both the generated visual quality and the semantic consistency**. This is an expected result, as the model's ability to extract the "prompt" (the abstract semantics) is impaired when the reference video itself is noisy or lacks detail.
>
> | Variant | Text CLIP Score $\uparrow$ | Motion Smoothness $\uparrow$ | Dynamic Degree $\uparrow$ | Aesthetic Quality $\uparrow$ | Semantic Alignment Score $\uparrow$ |
> | :--- | :---: | :---: | :---: | :---: | :---: |
> | **VAP + Noisy Ref (Gaussian, $\sigma$=0.05)** | 22.84 | 96.02 | 68.75 | 53.18 | 62.36 |
> | **VAP + Low-Res Ref (Downsampled)** | 22.95 | 95.48 | 66.67 | 56.60 | 65.52 |
> | **VAP (default)** | **24.13** | **98.59** | **77.08** | **57.71** | **70.44** |

---

> ### Author Response · Authors · 2025-11-19
> **Response to Reviewer 1zha (5/6)**
>
> > ### Question 2: Have the authors quantified improvements when using such synthetic instructions versus descriptive captions?
>
> We thank the reviewer for this question. This is a key point regarding the optimal use of our framework.
>
> 1.  **Rationale for Current Setup.** As stated in **Appendix E.1**, our current model adopts the **descriptive captioning style** of the base pretrained DiT. This was a deliberate choice to **preserve the base model’s generation capabilities**, which were trained on such captions. We agree that instruction-style captions are a powerful approach for controllable generation and have explicitly listed this as **future work** in the paper.
>
> 2.  **Annotation Costs.** We would like to clarify that **we have not conducted a full quantitative study** involving retraining VAP with instruction-style captions. Creating such annotations at scale and retraining the model is computationally significant and was **infeasible within the limited rebuttal period**.
>
> 3.  **Zero-Shot Evidence.** However, to provide preliminary evidence for this hypothesis, we *did* perform a **zero-shot test** using our current checkpoint (trained *only* on descriptive captions). We compared:
>
>     > * **Original Descriptive Caption:** "A chestnut-colored horse stands on a grassy hill against a backdrop of distant, snow-dusted mountains. The horse begins to inflate, its defined, muscular body swelling and rounding into a smooth, balloon-like form while retaining its rich, brown hide color. Without changing its orientation, the now-buoyant horse lifts silently from the ground... The camera remains completely static..."
>     > * **New "Weak" Instruction-Style Caption:** "The horse begins to inflate, become a balloon-like form, and lift silently from the ground."
>
>     Despite the checkpoint having **never seen such instruction-style captions** and despite these captions only mentioning the core semantics without describing object appearance or camera constraints, it **stably transferred the intended semantics** and achieved controllable generation [anonymous link: [https://anonymous.4open.science/r/Video-As-Prompt-Review-CA61/rebuttal/weak_instruction_example.mp4](https://anonymous.4open.science/r/Video-As-Prompt-Review-CA61/rebuttal/weak_instruction_example.mp4)]. While this is not a full quantitative comparison, it provides **strong initial evidence** that our framework is compatible with instruction-style captions, reinforcing this as a promising direction for future work.

---

> ### Author Response · Authors · 2025-11-19
> **Response to Reviewer 1zha (6/6)**
>
> > ### Question 3: Is the MoT design amenable to parameter-efficient variants (e.g., partial expert layers or shared attention blocks)? If so, did the authors explore such configurations?
>
> We thank the reviewer for this excellent question regarding parameter efficiency. We agree this is a valuable direction. Our explanation is as follows:
>
> 1.  **Design Focus (Expert-Level).** Our MoT design operates at the **expert level**, not by redesigning the internal DiT blocks. We introduce an **in-context transformer expert** (initialized from the pretrained DiT) to process the reference branch, while keeping the **original pretrained DiT frozen** to preserve its generative quality for the target branch. The two experts exchange information via **full attention** at corresponding layers.
>
> 2.  **Backbone-Agnostic Formulation.** Because our construction treats the DiT as a **black-box backbone**, the MoT formulation is compatible with **arbitrary internal DiT architectures**. In practice, we have instantiated and trained VAP with both **CogVideoX** and **Wan2.1** backbones, which empirically demonstrates that our MoT design is **backbone-agnostic and transferable**.
>
> 3.  **Parameter-Efficiency as Orthogonal Work.** Following from this, parameter-efficient variants (e.g., **partial expert layers, shared attention blocks, or other lightweight schemes**) primarily concern **modifications to the internal DiT architecture**, rather than the expert-level MoT formulation itself. While these directions are clearly important, they are **orthogonal to the main focus of this work**: establishing a **unified solution for semantic-controlled video generation** under the “video-as-prompt” paradigm. We therefore leave the detailed exploration of such parameter-efficient MoT/DiT variants to future work.

---

### Official Review · Reviewer_YyML · 2025-11-04

**Soundness:** 3
**Presentation:** 4
**Contribution:** 3
**Rating:** 8
**Confidence:** 4

**Summary:**

This paper addresses the significant challenge of achieving unified, generalizable semantic control in video generation. Current methods often fail by enforcing inappropriate pixel-wise priors from structure-based controls or by relying on non-generalizable, condition-specific finetuning and task-specific architectures.

The authors introduce Video-As-Prompt (VAP), a new paradigm that reframes this problem as in-context generation. VAP leverages a reference video as a direct semantic prompt. Its architecture augments a frozen base Video DiT with a plug-and-play MoT expert. This design prevents catastrophic forgetting, the trainable expert processes the reference prompt while the frozen backbone handles the target generation, with both communicating via full attention at each layer. A key component is a temporally biased RoPE, which breaks the model's default assumption of a pixel-aligned mapping between the prompt and the target video.

To train this model, the authors created and released VAP-Data, the largest dataset for this task, containing over 100,000 paired videos across 100 semantic conditions. As a single unified model, VAP sets a new state-of-the-art for open-source methods, achieves user preference rates comparable to leading commercial models, and demonstrates strong zero-shot generalization to unseen semantic concepts.

**Strengths:**

Unified and Generalizable Model: VAP is, to the authors' knowledge, the first framework to successfully unify a diverse set of semantic controls into a single model without requiring per-task modules or finetuning.

Strong Zero-Shot Performance: The model shows a strong ability to generalize to semantic conditions that were not included in the VAP-Data, such as "crumble" and "dissolve". This indicates it is learning a generalizable concept of semantic transfer.

VAP-Data: The paper introduces the largest-ever dataset for semantic-controlled video generation. This dataset, and the bootstrapping method used to create it, will be highly valuable for future research.

Ablation Studies: The paper provides a very strong set of ablations that convincingly justify the architectural design. The comparison clearly shows the superiority of the MoT design over finetuning, LoRA, unidirectional cross-attention, and residual addition. The ablation on RoPE design also confirms the authors' hypothesis .

**Weaknesses:**

he paper is strong and transparently discusses its own limitations.
Synthetic Data Limitations: The primary weakness, is that VAP-Data is entirely synthetic, generated using other models. The paper notes this means VAP may inherit the "stylistic biases, artifacts, and conceptual limitations" (e.g., bad hands) of these source models
Dependence on Caption Quality: Performance relies on well-aligned semantic descriptions in the reference and target captions. The authors show that mislabeled captions or a large mismatch in subject structure can significantly degrade generation quality.
Inference Cost: The plug-and-play MoT expert, while effective, adds significant computational overhead. The paper notes it roughly doubles inference time and increases memory usage.

**Questions:**

The multi-reference failure case suggests the model struggles to disentangle pure semantics from appearance/layout when the reference videos are structurally diverse, leading to "leakage" of features like "spider legs" and "fish shape". You hypothesize this stems from using generic video captions. Do you believe training with more explicit "instruction-style" captions would be sufficient to solve this, or does this failure point to a need for a more explicit architectural component to enforce disentanglement?
Regarding the synthetic VAP-Data, your zero-shot results are promising. However, one could argue the model is learning a "meta-task" of how to follow VFX-style instructions from the synthetic data, rather than a truly general semantic understanding. How does VAP perform on zero-shot tasks that are semantically distinct from the VFX-style domain?
The temporally biased RoPE is a key innovation. You mention shifting the reference's temporal indices by a fixed offset $\Delta$. How was this offset value chosen, and how sensitive is the model's performance to this hyperparameter? Furthermore, does the MoT expert learn to attend to the entire reference video prompt equally at each step of the target generation, or does it learn a dynamic temporal correspondence?

---

> ### Author Response · Authors · 2025-11-19
> **Response to Reviewer YyML (1/7)**
>
> We express our sincere gratitude to the reviewer for the insightful comments and the recognition of our work. We especially appreciate the acknowledgment of our approach's novelty, effectiveness, and superior performance. We have addressed the specific concerns raised by the reviewer as detailed below.
>
>
>
> > ### Weakness 1: Synthetic Data Limitations
>
>
> We thank the reviewer for this important point. As discussed in Sec. 5 Conclusion (Limitation and Future Works) and Appendix D Dataset, we chose synthetic data construction because creating **a large-scale, real-world, semantically controlled video dataset** is *infeasible* at this exploratory stage due to **extremely high annotation costs**.
>
> However, we believe our VAP-Data and the Video-As-Prompt framework offer significant contributions to the community in three key areas:
>
>
> 1. **Data Contribution.** Although VAP-Data is synthetic, it is, to the best of our knowledge, the **largest available dataset** for semantic control in video generation. The previous largest dataset, **Open-VFX** ([https://huggingface.co/datasets/sophiaa/Open-VFX](https://huggingface.co/datasets/sophiaa/Open-VFX)), is also synthetic and contains only 559 samples across 14 semantic conditions. In contrast, VAP-Data expands the scale by over **178×, offering more than 100k videos across 100 semantic conditions**. This provides **the first practical, training-friendly data foundation** for *unified* semantic-controlled video generation.
>
> 2.  **Research Contribution.** Our primary goal is to study **how to transfer abstract semantics from a reference video to a target video** under a unified formulation. Our *Video-As-Prompt* approach, combined with the MoT expert and temporally-biased RoPE, validates this core idea. To our knowledge, this is the **first work in the community to validate the "video-as-prompt" concept for unified semantic control**, filling an important gap in controllable video generation.
>
> 3.  **Generalization to Real-World Data.** To further assess generalization, we **added zero-shot experiments** to show that although VAP is trained solely on synthetic data, it **generalizes well to real-world reference videos and complex composite semantics**. For example, our model can reproduce:
>
>       * The complex semantics of a character raising both arms with their back to an explosion (e.g., the "Iron Man" movie) [anonymous link: [https://anonymous.4open.science/r/Video-As-Prompt-Review-CA61/rebuttal/real\_video\_example\_1.mp4](https://anonymous.4open.science/r/Video-As-Prompt-Review-CA61/rebuttal/real_video_example_1.mp4)].
>       * The famous real-world "Michael Bay shot" (a rapid, low-angle, orbiting shot) [anonymous link: [https://anonymous.4open.science/r/Video-As-Prompt-Review-CA61/rebuttal/real\_video\_example\_2.mp4](https://anonymous.4open.science/r/Video-As-Prompt-Review-CA61/rebuttal/real_video_example_2.mp4)].
>       * Complex composite effects, such as transforming into a robot (i.e., "Transformer") while the environment simultaneously turns to winter [anonymous link: [https://anonymous.4open.science/r/Video-As-Prompt-Review-CA61/rebuttal/multi\_vfx\_example.mp4](https://anonymous.4open.science/r/Video-As-Prompt-Review-CA61/rebuttal/real_video_example_3.mp4)].
>
> Finally, we sincerely agree with the reviewer on the importance of a large-scale, real-world, semantic-controlled video dataset. As stated in our limitations (Line 480), we leave this as an important direction for future work.

---

> ### Author Response · Authors · 2025-11-19
> **Response to Reviewer YyML (2/7)**
>
> > ### Weakness 2: Dependence on Caption Quality
>
> We thank the reviewer for highlighting this important point. We explain this in three parts:
>
> 1.  **Source of Caption Dependence.** As discussed in **Appendix E.1**, this limitation primarily stems from our decision to **follow the original pretraining caption distribution of the base Video DiTs**. These models are pretrained with long, descriptive captions to ensure high-quality generation. Consequently, if the captions are noisy, misaligned, or of low quality, the **base Video DiT itself suffers a degradation in generation quality**. This is consistent with community observations.
>
> 2.  **Role of Captions in Video-As-Prompt.** We empirically observe that Video-As-Prompt first relies on the **most critical shared semantics** in the reference and target captions to establish a coarse-level "anchor" for the concept to be transferred. It then leverages the **reference video** to fill in finer-grained details, such as stylistic attributes, motion patterns, or camera speed. When captions are vague or incorrect, this semantic anchor becomes ambiguous, which in turn **impairs the model's semantic selection** and leads to a performance drop.
>
>
> 3.  **Potential Optimization.** As stated in **Sec. 5: Limitations and Future Works (lines 481–485)**, we agree this is a key area for improvement. We hypothesize that **instruction-style captions** (e.g., *“please follow the Ghibli style in the reference video”*) would better capture the **intended semantic signal**, even when objects differ explicitly between reference and target. This would likely improve robustness for practical applications. We leave this promising direction to future work due to the significant instruction annotation and model retraining costs involved.
>
>
> 4.  **Zero-Shot Evidence with Instruction-Style Captions.** To provide preliminary evidence for this hypothesis, we performed a **zero-shot test** using our current checkpoint (which was trained *only* on descriptive captions). We compared the output using:
>
>     * **Original Descriptive Caption:** "A chestnut-colored horse stands on a grassy hill against a backdrop of distant, snow-dusted mountains. The horse begins to inflate, its defined, muscular body swelling and rounding into a smooth, balloon-like form while retaining its rich, brown hide color. Without changing its orientation, the now-buoyant horse lifts silently from the ground. ... The camera remains completely static..."
>
>     * **New "Weak" Instruction-Style Caption:** "The horse begins to inflate, become a balloon-like form, and lift silently from the ground."
>
>     Despite the checkpoint having **never seen such instruction-style captions during training**—and despite these captions only mentioning the core semantics without describing object appearance or camera constraints—we observed that the model could still **stably transfer the intended semantics and achieve controllable generation** (anonymous link: [https://anonymous.4open.science/r/Video-As-Prompt-Review-CA61/rebuttal/weak_instruction_example.mp4](https://anonymous.4open.science/r/Video-As-Prompt-Review-CA61/rebuttal/weak_instruction_example.mp4)). While this is not a full quantitative comparison, it provides **strong initial evidence** that instruction-style captions are compatible with our framework and represent a promising direction for future, systematic study.

---

> ### Author Response · Authors · 2025-11-19
> **Response to Reviewer YyML (3/7)**
>
> > ### Weakness 3: Inference Cost
>
>
> We thank the reviewer for this question and would like to clarify the distinction between **computation (inference time)** and **memory**.
>
> 1.  **Computation vs. Memory.** We appreciate the concern and would like to clarify that the inference cost of VAP can be decomposed into **compute (which determines inference time)** and **memory**. Our adopted MoT architecture primarily increases **memory usage** by introducing an additional in-context DiT expert. However, in terms of **compute**, our approach is equivalent to a baseline that directly extends the original DiT input to (reference_caption, reference_video, target_caption, target_video) and performs standard in-context generation.
>
> 2.  **MoT vs. Single-DiT Compute.** To be precise, for in-context generation, the dominant computation comes from the **joint full attention over all reference and target video tokens and the feedforward (FFN) layers**. In our MoT, the attention mechanism is still full attention, and the FFN computation is simply split across two experts with separate parameters. This results in the **same overall FLOPs** as using a single, unified DiT with one shared FFN module applied to all tokens. In other words, **the MoT structure does not increase compute** compared to a single-DiT in-context setup; it primarily changes the parameterization and memory footprint.
>
> 3.  **Source of Latency and Revision.** Therefore, the observed **increase in inference time (roughly 2×) stems from the in-context generation setup itself** (i.e., processing both reference and target data), **not from the MoT architecture.** We have revised our manuscript to correct our original, imprecise wording. We changed the text from:
>     > “Inference time roughly doubles on average, mainly due to additional MoT-expert computation and in-context full attention”
>
>     to:
>
>     > “Inference time roughly doubles on average, mainly due to **additional in-context computation**.”
>
> 4.  **Parameter-Efficient Variants and Scope.** We also *have* explored **more parameter- and memory-friendly variants** of the in-context expert in **Appendix F (Ablation Study)**, specifically in **Table 7** (*In-Context Expert Transformer Layer Distribution*).
>     * While **fully copying the pre-trained DiT** as the in-context expert (adopted by our CogVideoX version) achieves the strongest performance, we show that **distributed / interval replication** of the expert layers (adopted by our Wan2.1 version) delivers competitive control while **reducing memory consumption and the number of layers performing full in-context attention** (thus lowering effective compute).
>     * A more systematic exploration of **parameter- and memory-efficient MoT designs** (e.g., more aggressive sharing, pruning, or sparse attention) is **orthogonal to our core contribution** of establishing a framework for unified semantic control. As discussed in Appendix E.3 (Efficiency), we explicitly leave these efficiency-focused optimizations to future work.

---

> ### Author Response · Authors · 2025-11-19
> **Response to Reviewer YyML (4/7)**
>
> > ### Question 1: Multi-Reference Failure & Disentanglement
>
>
>
> We thank the reviewer for this insightful question. We agree and believe that **training with more explicit, instruction-style captions would largely address this issue**, likely without requiring new architectural components for explicit disentanglement.
>
> 1.  **Cause of the Failure Case.** As discussed in **Lines 1213–1221**, the showcased multi-reference failure case arises because the provided reference videos **differ significantly in both structure and their semantic realization** of the target semantic:
>     * The references mix disparate shapes (e.g., **human, spider, and flatfish**).
>     * Semantically, they are inconsistent: Reference 1 clearly depicts "AI Lover Drop"; Reference 2 shows a falling spider without a hug; Reference 3 is weakest, showing a flatfish swimming upward.
>     * Collectively, these references **fail to express a unified concept** (e.g., “a lover of the same species falling from the sky”), and their captions do not provide this shared semantic context. This mismatch causes the model to **blend incompatible appearance and layout cues**, leading to the observed entanglement (e.g., spider legs, fish shape).
>
> 2.  **Impact of Instruction-Style Captions.** We hypothesize that **instruction-based captions** would resolve this ambiguity. An explicit instruction (e.g., *“Mimic these scenarios: generate a lover of the same species who falls from the sky”*) would provide a strong, clear semantic anchor across all references. We expect this would **significantly reduce ambiguity and the blending of conflicting features**, framing this as a **data-format problem** rather than one requiring a structural redesign for disentanglement.
>
> 3.  **Scope of this Experiment.** Finally, we wish to clarify that our core contribution is **unified semantic-controlled video generation** under the “video-as-prompt” paradigm. The multi-reference setting was presented as an **exploratory experiment** to show potential future directions, not as a fully developed module. A **comprehensive study of multi-reference training and disentanglement** is beyond the scope of this work and is explicitly left for future research.

---

> ### Author Response · Authors · 2025-11-19
> **Response to Reviewer YyML (5/7)**
>
> > ### Question 2: Zero-shot Performance Out of the VFX-style Domain
>
> We thank the reviewer for this important question regarding generalization beyond our synthetic training domain. To explicitly address this, we have added new experiments and would also like to clarify the unique zero-shot capabilities of our framework.
>
> 1.  **New Real-World Zero-Shot Evaluation.** To further assess generalization, we have **added new zero-shot experiments**. These results demonstrate that although VAP is trained solely on synthetic data, it **generalizes well to real-world reference videos and complex composite semantics**. For example, our model can successfully reproduce:
>
>       * The complex semantics of a character raising both arms with their back to an explosion (e.g., the "Iron Man" movie) [anonymous link: [https://anonymous.4open.science/r/Video-As-Prompt-Review-CA61/rebuttal/real\_video\_example\_1.mp4](https://anonymous.4open.science/r/Video-As-Prompt-Review-CA61/rebuttal/real_video_example_1.mp4)].
>       * The famous real-world "Michael Bay shot" (a rapid, low-angle, orbiting shot) [anonymous link: [https://anonymous.4open.science/r/Video-As-Prompt-Review-CA61/rebuttal/real\_video\_example\_2.mp4](https://anonymous.4open.science/r/Video-As-Prompt-Review-CA61/rebuttal/real_video_example_2.mp4)].
>       * Complex composite effects, such as transforming into a robot (i.e., "Transformer") while the environment simultaneously turns to winter [anonymous link: [https://anonymous.4open.science/r/Video-As-Prompt-Review-CA61/rebuttal/multi\_vfx\_example.mp4](https://anonymous.4open.science/r/Video-As-Prompt-Review-CA61/rebuttal/real_video_example_3.mp4)].
>
>
> 2.  **Clarification on the Novelty of our Zero-Shot Capability.** We would also like to respectfully highlight the novelty of the zero-shot evaluation *already present* in our original submission, as detailed in **Figure 7**, **Lines 192–196**, and **Lines 410–413**. The semantic conditions shown there are **never seen during training**, making the results a **genuine zero-shot generalization**. To the best of our knowledge, existing methods (including commercial APIs) rely on **condition-specific or task-specific finetuning**. They **do not support zero-shot semantic generalization** to unseen effects within a *single unified model*. In contrast, **Video-As-Prompt** is, to our knowledge, the **first unified framework** to achieve this by treating the reference video as a prompt.

---

> ### Author Response · Authors · 2025-11-19
> **Response to Reviewer YyML (6/7)**
>
> > ### Question 3: Choice and Sensitivity of the Temporally Biased RoPE Offset
>
> We thank the reviewer for this constructive question. We have added a detailed explanation of our design principles and a new ablation study to demonstrate the robustness of our approach to this hyperparameter.
>
> 1.  **Design Principle.** Our choice of the temporal offset ($\Delta$) is governed by the core principle of **avoiding temporal overlap between reference and target videos while keeping their spatial positions aligned**. In our default setup, RoPE is applied only to video tokens. Let the temporal length of both the reference and target videos be $T$. We assign the target video temporal indices $[0, T-1]$ and set $\Delta = T$, so the reference video uses the indices $[-T, -1]$. This temporally biased RoPE:
>     * Prevents any artificial one-to-one pixel mapping prior between reference and target frames.
>     * Intuitively aligns with the concept of **in-context video generation**, where the reference serves as a context rather than as frame-aligned supervision.
>
>     As shown in **Table 2 (Position Embedding Design)** and **Fig. 5(c,d)**, this design empirically yields superior performance by avoiding spurious mapping priors.
>
> 2.  **Hyperparameter Choice and Ablations.** We further clarify the choice of $\Delta$ and have added new ablations on its sensitivity. We tested four variants against our default:
>     * **A1 (larger, multiple of $T$):** $\Delta = 2T$
>     * **A2 (larger, non-multiple):** $\Delta = 1.5T$
>     * **A3 (smaller, partial overlap):** $\Delta = 0.5T$
>     * **A4 (opposite sign):** $\Delta = -T$ (placing the reference *after* the target)
>
>     As shown in the table below, we found that the model is **generally insensitive** to the exact value of $\Delta$, as long as the temporal position ranges of the reference and target videos do not overlap. The quantitative metrics for A1, A2, and A4 remain nearly identical to our default.
>
> 3.  **Robustness and Takeaway.** The only case with a clear performance degradation is **A3 ($\Delta = 0.5T$)**. Here, the temporal ranges partially overlap, which reintroduces the erroneous priors that implicitly encourage a non-existent pixel-wise correspondence, thus hurting performance. These results indicate that the model is **not highly sensitive to the exact value of $\Delta$**. The key requirement is simply that the **reference and target temporal positions remain disjoint** to prevent injecting an artificial alignment prior.
>
>
> | Variant | Text CLIP Score $\uparrow$ | Motion Smoothness $\uparrow$ | Dynamic Degree $\uparrow$ | Aesthetic Quality $\uparrow$ | Semantic Alignment Score $\uparrow$ |
> | :--- | :---: | :---: | :---: | :---: | :---: |
> | **A1 ($\Delta = 2T$)** | 24.05 | 98.63 | 75.00 | 57.68 | 70.23 |
> | **A2 ($\Delta = 1.5T$)** | 24.07 | 98.57 | 79.17 | 57.62 | 70.41 |
> | **A3 ($\Delta = 0.5T$, overlap)** | 23.62 | 98.52 | 72.92 | 57.25 | 69.24 |
> | **A4 ($\Delta = -T$)** | 24.03 | 98.55 | 77.08 | 57.70 | 70.15 |
> | **VAP ($\Delta = T$, default)** | **24.13** | **98.59** | **77.08** | **57.71** | **70.44** |

---

> ### Author Response · Authors · 2025-11-19
> **Response to Reviewer YyML (7/7)**
>
> > ### Question 4: Does the MoT expert learn to attend to the entire reference video prompt equally at each step of the target generation, or does it learn a dynamic temporal correspondence?
>
>
> Thank you for this excellent question. We confirm that the MoT expert **learns a dynamic temporal correspondence** between the reference video prompt and the target video, which evolves at different stages of the denoising process.
>
> 1.  **Empirical Analysis.** To demonstrate this, we analyzed our CogVideoX-based Video-As-Prompt implementation with **50 inference steps**. We selected a reference-target video pair [anonymous link: [https://anonymous.4open.science/r/Video-As-Prompt-Review-CA61/rebuttal/weak_instruction_example.mp4](https://anonymous.4open.science/r/Video-As-Prompt-Review-CA61/rebuttal/weak_instruction_example.mp4)] and visualized the average attention maps from the target video tokens to the reference video tokens at denoising steps **1, 25, and 50**. (The visualization is at the frame level, where all tokens for a given frame are averaged into a single representation).
>
> 2.  **Observed Behavior.** We observe a distinct, three-stage pattern [anonymous link: [https://anonymous.4open.science/r/Video-As-Prompt-Review-CA61/rebuttal/denosing_process.png](https://anonymous.4open.science/r/Video-As-Prompt-Review-CA61/rebuttal/denosing\_process.png)]:
>     * **Step 1 (Early Denoising):** The target frames initially pay more attention to **temporally adjacent** reference frames, suggesting a simple, localized alignment.
>     * **Step 25 (Mid-Denoising):** As denoising progresses, the attention spreads, becoming more **broadly and uniformly** focused on the general temporal region where the core semantics occur (e.g., the first half and the last few frames).
>     * **Step 50 (Late Denoising):** As the generation is finalized, the attention becomes highly **focused on specific semantic details** (e.g., the central and ending frames where the action is most pronounced).
>
>     This progression clearly indicates that the MoT expert **gradually refines the temporal correspondence** throughout the generation process, rather than using a fixed, uniform attention pattern.

---

### Author Response · Authors · 2025-11-19
**General Response to Common Questions**

We sincerely thank all the reviewers for your constructive feedback and recognition of this work. We are encouraged that reviewers acknowledged 1) our **novel "video-as-prompt" reformulation** (all reviewers), 2) the **significant contribution of the first unified semantic-controlled framework** (all reviewers), 3) the **elegant design of MoT and Temporally-Biased RoPE** (all reviewers), 4) the **foundational value of the largest semantic-controlled video generation dataset VAP-Data** (Reviewers YyML, 1zha, sfsG), 5) the **clear and good presentation** (Reviewer YyML, 1zha, v4tD), and 6) the **comprehensive evaluation and impressive visual results** (Reviewer YyML, 1zha, sfsG, v4tD).

We have carefully addressed all suggestions and incorporated extensive experiments, with the needed modified content highlighted in blue in the revised manuscript. Below is a summary of our key responses and improvements:

  * **Generalization to Real-World Data & Zero-shot Evaluation** (Reviewers YyML, 1zha, v4tD, sfsG)

      * **New Zero-shot Experiments:** We added **new zero-shot experiments** on challenging real-world reference videos (e.g., "Iron Man" explosion ending pose, "Michael Bay" shot) to further demonstrate strong out-of-domain generalization.
      * **New Quantitative Evidence:** We added **quantitative evaluations** for current zero-shot settings, showing that performance metrics on unseen semantics remain consistent with seen semantics.
      * **Novelty:** We clarified that VAP is the first unified framework to achieve this semantic transfer to unseen effects **without condition (task)-specific finetuning**, distinguishing it from existing commercial tools and open-source baselines.

  * **Synthetic Data Limitations & Dataset Contribution** (Reviewers YyML, 1zha, v4tD)

      * **Justification & Scale:** We clarified that constructing large-scale real-world semantic datasets is currently infeasible due to annotation costs. We emphasized that **VAP-Data** is the largest available foundation, expanding the scale by **178×** over the previous largest dataset (Open-VFX).
      * **Validation of Contribution:** We argued that our core contribution is validating the "video-as-prompt" paradigm for unified control. Despite being trained on synthetic data, our extensive experiments prove the model effectively extracts abstract semantics and **generalizes to real-world scenarios**, bridging the domain gap.


  * **Computational Cost & Model Efficiency** (Reviewers YyML, 1zha, KiRJ, sfsG)

      * **Compute vs. Memory:** We clarified that the MoT overhead is primarily **memory** (parameter count), while the increased inference time is inherent to the **in-context generation task** itself, not the MoT architecture.
      * **Efficiency Ablations:** We expanded the analysis in **Appendix F**, exploring parameter-efficient variants (e.g., interval expert replication). Results show that lightweight designs can also achieve competitive performance with significantly reduced memory footprints.

  * **Robustness to Caption Quality & Instruction-Style Prompts** (Reviewers YyML, 1zha, v4tD)

      * **Mechanism Analysis:** We analyzed the model's reliance on descriptive captions, explaining it stems from preserving the base model's pretraining distribution.
      * **Promising Zero-shot Instruction-style Caption:** We provided **preliminary evidence** via a zero-shot test where the model successfully followed "weak" instruction-style captions (e.g., *"The horse begins to inflate..."*) despite no explicit training. This suggests strong potential for future instruction tuning.
      * **Failure Analysis:** We extended the failure case analysis to explain how ambiguous semantic anchors in noisy captions can confuse the selection mechanism.

  * **Theoretical Analysis & Mechanism Interpretation** (Reviewers YyML, KiRJ)

      * **Theoretical Formulation:** We added a formal analysis to explain how the in-context MoT expert effectively disentangles and transfers semantic patterns via **content-based attention matching**.
      * **Generation Process Visualization:** We visualized the **dynamic temporal correspondence** learned by the MoT expert, revealing a coarse-to-fine attention evolution (global context $\to$ specific details) during the denoising process.
      * **RoPE Ablation:** We added ablation studies on the **Temporally Biased RoPE offset ($\Delta$)**, demonstrating the model's robustness as long as temporal overlap is avoided.

  * **Additional Comparisons & Reproducibility** (Reviewers sfsG, v4tD)

      * **Comparison with Omni-Effects:** We added a **quantitative comparison** with the concurrent Omni-Effects, demonstrating our method's superior generalization and unified nature compared to concurrent LoRA-MoE-based approaches.
      * **Open Source Commitment:** We reaffirmed our commitment to releasing the full **VAP-Data dataset**, code (training and inference), and checkpoints to foster future research.

---

### Author Response · Authors · 2025-12-02
**Summary of Rebuttal and Report of an Incorrectly Uploaded Review**

**Dear Area Chair**,

**We briefly summarize our rebuttal**: we added new real-world and zero-shot experiments (with qualitative visualization and quantitative evaluation), efficiency ablations, robustness analyses, and a clearer justification of VAP-Data and our unified “video-as-prompt” contribution.

**We also respectfully report an incorrectly uploaded review.** The review by Reviewer sfsG now on our page (score 2, reject) appears to be about another paper: [https://openreview.net/forum?id=8FihPljvWf&noteId=4BWbnYsXpt](https://openreview.net/forum?id=8FihPljvWf&noteId=4BWbnYsXpt). The review on that paper by Reviewer aNEz (score 6, “marginally above the acceptance threshold”) appears to be about ours: [https://openreview.net/forum?id=vYqAuAuV1v&noteId=OLijiGXsUc](https://openreview.net/forum?id=vYqAuAuV1v&noteId=OLijiGXsUc). We kindly ask you to remove or publicly annotate the incorrect review to keep the evaluation accurate and avoid misleading other reviewers and readers.

**Sincerely**,

**Video-As-Prompt Authors**

---

### Meta-Review · Area_Chair_2Wza · 2026-01-07

**Summary:**

Reviewers' concerns:
* Synthetic data limitations. VAP-Data is fully synthetic and may bias the model toward VFX-style semantics rather than general understanding.
* Efficiency and scalability. The MOT design introduces an additional expert transformer alongside a frozen backbone, which roughly doubles inference latency and increases memory consumption.
* Sensitivity to caption quality. The performance relies on well-aligned semantic descriptions in the reference and target captions. Reviewers were concerned that in realistic scenarios, captions may be noisy or partially incorrect, which could cause the model to produce degraded or misleading generations.
* Lack of a theoretical analysis explaining why in-context learning via Mixture-of-Transformers effectively transfers semantic patterns. Similarly, the proposed temporally biased RoPE is only justified empirically, without an ablation or analytical study on the optimal bias magnitude.

The paper is generally viewed as sound, well-motivated, and performs well. The authors’ rebuttal addressed most major concerns, pushing the balance toward acceptance.

**Reviewer Concerns:**

Concerns largely addressed by the rebuttal:

* Lack of theoretical explanation. The authors added a formal attention-based interpretation and explained the role of frozen backbone + expert.

Concerns Partially Addressed:

* Synthetic data limitations. Zero-shot demos help, but no systematic real-data training or evaluation.
* Efficiency and scalability. Clarified that compute overhead is inherent to in-context generation, not MoT itself. MoT mainly increases memory, so the memory efficiency issue remains. Added parameter-efficient variants with competitive performance. No deep exploration of highly optimized or sparse designs.
* Sensitivity to caption quality. Failure analysis and explanation are added. The authors also provided qualitative zero-shot evidence for instruction-style captions. But there's no quantitative study or retraining with instruction-style captions.

**Reviewer Scores:**

Reviewer YyML

Original score: 8 (Accept)

Likely change: 8 (unchanged)

Reason: This reviewer already rated the paper highly. The rebuttal confirms the positive assessment rather than changing it.

Reviewer 1zha

Original score: 6 (Marginally above acceptance threshold)

Likely change: 6 or 8

Reason: The concerns on synthetic data, efficiency, and caption dependence are partially addressed.

Reviewer KiRJ

Original score: 6 (Marginally above acceptance threshold)

Likely change: 8

Reason: The rebuttal provides a formal attention-based interpretation and RoPE offset ablations, which well addressed the concern on theoretical explanation.

Reviewer sfsG is a misassigned reviewer. Its score should not be counted.

Reviewer v4tD

Original score: 6 (Marginally above acceptance threshold)

Likely change: 6 or 8

Reason: The concerns on synthetic data, zero-short evaluation, and clarity are partially addressed.

---

### Decision · Program_Chairs · 2026-01-26

Accept (Poster)